# WindsorML: High-Fidelity Computational Fluid Dynamics Dataset For Automotive Aerodynamics

**Neil Ashton**[*]
Amazon Web Services
60 Holborn Viaduct
London, EC1A 2FD

**Jordan B. Angel**
Volcano Platforms Inc.
Stanford Research Park, 3240 Hillview Ave
Palo Alto, CA 94304

**Aditya S. Ghate**
Volcano Platforms Inc.
Stanford Research Park, 3240 Hillview Ave
Palo Alto, CA 94304

**Gaetan K. W. Kenway**
Volcano Platforms Inc.
Stanford Research Park, 3240 Hillview Ave
Palo Alto, CA 94304

**Man Long Wong**
Volcano Platforms Inc.
Stanford Research Park, 3240 Hillview Ave
Palo Alto, CA 94304

**Cetin Kiris**
Volcano Platforms Inc.
Stanford Research Park, 3240 Hillview Ave
Palo Alto, CA 94304

**Astrid Walle**
Siemens Energy
Huttenstraße 12, 10553
Berlin, Germany

**Danielle C. Maddix**
AWS AI Labs
2795 Augustine Dr.,
Santa Clara, CA 95054, United States

**Gary Page**
Loughborough University
Epinal Way, Loughborough
LE11 3TU, United Kingdom

## Abstract

This paper presents a new open-source high-fidelity dataset for Machine Learning (ML) containing 355 geometric variants of the Windsor body, to help the development and testing of ML surrogate models for external automotive aerodynamics. Each Computational Fluid Dynamics (CFD) simulation was run with a GPU-native high-fidelity Wall-Modeled Large-Eddy Simulations (WMLES) using a Cartesian immersed-boundary method using more than 280M cells to ensure the greatest possible accuracy. The dataset contains geometry variants that exhibits a wide range of flow characteristics that are representative of those observed on road-cars. The dataset itself contains the 3D time-averaged volume & boundary data as well as the geometry and force & moment coefficients. This paper discusses the validation of the underlying CFD methods as well as contents and structure of the dataset. To the authors knowledge, this represents the first, large-scale high-fidelity CFD dataset for the Windsor body with a permissive open-source license (CC-BY-SA).

## 1   Introduction

The use of Machine Learning (ML) to augment Computational Fluid Dynamics (CFD) is gaining the attention of academia and industry because of its potential to offer an additional tool to explore new designs in near real-time compared to traditional CFD simulations (1; 2; 3; 4; 5). Whilst these methods are still in their infancy, there have been promising results that show their ability to

---

[*]Now at NVIDIA, Corresponding Author: nashton@nvidia.com , contact@caemldatasets.org

38th Conference on Neural Information Processing Systems (NeurIPS 2024) Track on Datasets and Benchmarks.

predict surface, volume and/or forces and moments on unseen geometries or boundary conditions (6; 7; 1; 8; 9). However the majority of these examples have been for 2D cases (6) with limited examples on more complex 3D cases (1; 9; 7; 8). One of the reasons for the limited dissemination of 3D examples is the lack of publicly available 3D training data. Unlike Large-Language Models (LLMs) where the training data is readily available, public training data for CFD, based upon 3D realistic geometries is limited and where such examples exist they are often using lower-fidelity CFD methods (8; 6) that are not representative of industry state of the art. Generating such datasets also requires a combination of large-scale High-Performance Computing (HPC) resources, human time to build an efficient workflow and in-depth knowledge of the underlying test-case to ensure rigour and correlation to the corresponding experimental data.

To help to address this lack of 3D training data, Ashton et al. (10) recently created the AhmedML dataset, that contains 500 different geometry variants of the Ahmed car body, simulated using a high-fidelity hybrid RANS-LES approach using OpenFOAM on meshes of approximately 20M cells. This open-source dataset contains time-averaged volume & boundary flow-field variables as well as forces and moment data to provide a rich dataset to aid ML model development. Whilst this dataset enables the development and testing of ML methods, there are a number of limitations to this dataset that have motivated the creation of the WindsorML dataset.

Firstly, in order to thoroughly assess an ML method, more than one dataset is required. Just as traditional CFD approaches are validated on a number of cases to ensure general applicability. Secondly, the AhmedML dataset is based upon mesh sizes of approximately 20M cells which is below the industry standard of $> 100$M cells for scale-resolving simulations of road-cars (11) and limits the ability to test out the scaling of ML methods using realistic mesh sizes. This is an important point as some ML methods that work well for cases at 20M cells may become inefficient as you scale beyond 300M points (1) and/or require the use of downsampling and data reduction techniques. Moreover from a flow physics point of view, the Windsor body (12; 13) is closer to modern road-cars compared to the Ahmed car body and contains more detailed and up to date experimental data e.g tomographic 3D PIV data.

Finally, the choice of the Windsor body is also motivated by its use within the Automotive CFD Prediction Workshops (AutoCFD) [2] that aim to bring the fluid dynamics and automotive community together to improve the state of the art for CFD prediction of road vehicles. The dataset described in this paper was used as a test-case during the 4th AutoCFD workshop, for which future papers will discuss the resulting comparisons from a range of different ML methods. Concurrently the DrivAerML dataset (14) based upon the open-source Drivaer has been created, that is described in it's own paper. Used together, the AhmedML (10), WindsorML and DrivAerML (14) datasets, provide ML developers with a broad set of data to use that has been formatted in a consistent fashion and is openly available to download and use with a permissive license (CC-BY-SA). A dedicated website [3] for these datasets has been created to provide the community with up to date information.

## 1.1  Main Contributions

This paper's novel contributions are summarized as follows:

- 355 variations in the Windsor body geometry that cover a broad range of pressure and geometry induced flow separation;

- the use of high-fidelity wall-modeled Large-Eddy Simulation (WMLES) CFD method which ensures the best possible correlation to the ground truth;

- first ever, freely available, open-source (CC-BY-SA) large-scale dataset based upon the Windsor body that can be used to train ML methods for automotive aerodynamics use cases.

The paper is organised as follows: first, the Windsor test-case is described together with the available experimental data. Next, the validation of the baseline geometry and test-case are described. The results section focuses on both global force coefficients and also selected off-body quantities such as planes in the wake of the car. Next, the dataset itself is detailed, including the choice of the geometry variations and the specific outputs that are included in the dataset. Finally, conclusions are drawn

---

[2]https://autocfd.org
[3]https://caemldatasets.org

and future work is described. The Supplementary Information (SI) contains additional details on the dataset and validation.

## 2   Test-Case Description

The Windsor body was created by Steve Windsor and Jeff Howell of Rover and later Jaguar Land Rover (JLR) (12; 13) to improve upon the well studied Ahmed car body (15; 16; 17; 18) by creating a geometry whose flow features more closely match a road vehicle. Whilst the Ahmed car body has been used in many experimental and CFD studies, the long flat surface between the front and rear of the body decouples the aerodynamic behaviour of these two regions and makes it more relevant to commercial vehicles. Two variants of the Windsor body were experimentally tested at the Loughborough University wind-tunnel (12) and have been the subject of numerous CFD studies - that most recently have taken place within the 2nd, 3rd and 4th Automotive CFD Prediction Workshops (13). The Windsor body was tested in numerous configurations but for this study we follow the 4th AutoCFD4 setup, where the vehicle has no wheels (which simplifies a common source of uncertainty) and the squareback rear shape is used, as the baseline geometry. As discussed later, this baseline geometry is then adapted according to 7 parameters to generate 355 different geometries.

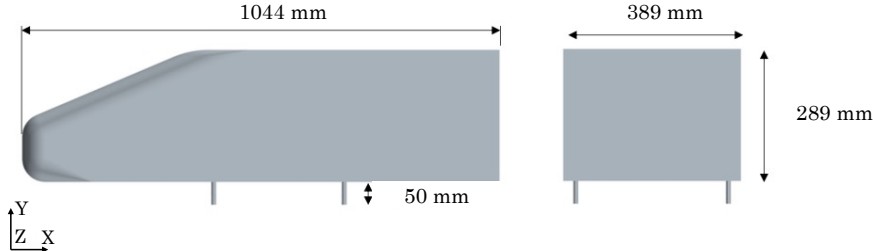

Figure 1: Windsor Geometry

### 2.1   Geometry

The model geometry is shown in 1. The reference frontal area is defined by the vehicle height and width and for the baseline geometry is $0.112\,\mathrm{m^2}$. The reference length used for pitching moment is the wheelbase $0.6375\,\mathrm{m}$. The CAD geometry of the model has its origin on the ground plane, in the symmetry plane midway between the wheels (which are not included in this particular geometry) i.e $x, y, z = (0, 0, 0)$. This is also the moment reference center for the calculating of the moments.

The model is yawed by $-2.5°$ around the $y$-axis (please note that for all the simulations in this work, $y$ is upwards, which differs to the original Windsor geometry where $z$ is upwards) , so generating a positive side force consistent with the experiment. The experimental forces and moments are in the coordinate system of the yawed model, whilst the PIV measurements are in the wind tunnel coordinate system. The choice of $-2.5°$ yaw was to avoid the well documented bi-stability that can often occur on these bluff bodies (19) that would cause unnecessary uncertainties (but nevertheless deserves future investigation).

The model is mounted in the wind tunnel with four pins at a ground clearance of $50\,\mathrm{mm}$ and zero pitch. In order to more closely match the underlying experimental data, the simulations themselves include a simplified wind tunnel domain i.e. $3.2\,\mathrm{m}$ long working section with a $1.92\,\mathrm{m} \times 1.32\,\mathrm{m}$ (width by height) cross section. There is no moving ground plane so a boundary layer grows along the groundplane. Experimental measurements (12) at the centre of the working section quote a boundary layer thickness of $60\,\mathrm{mm}$, displacement thickness of $9.4\,\mathrm{mm}$ and momentum thickness of $5.5\,\mathrm{mm}$. The maximum turbulence intensity was measured to be approximately 3% at the edge of the boundary layer.

### 2.2   Solver Setup

All simulations were run using the Volcano ScaLES code by Volcano Platforms, which solves the explicit compressible Navier–Stokes equations using a nominally 4th order spatially accurate

finite difference discretization with favorable Kinetic Energy and Entropy Consistency properties making it suitable for Large Eddy Simulations (LES) over a range of flow regimes. The viscous flux discretization utilizes a mix of 4th order and 2nd order discretizations with high spectral bandwidth thereby allowing further robustness for high Reynolds number flows without additional numerical dissipation via the inviscid flux numerical operator in turbulence resolving regions. Physics-based numerical sensors that are functions of local velocity gradients as well as pressure and density fluctuations allow for spatio-temporally localized use of limiters needed to capture flows with discontinuities such as shocks and steep density gradients. The Strong Stability Preserving (SSP) 3rd order Runge–Kutta scheme developed by Gottlieb & Shu (20) is utilized for all the work presented in this paper.

Geometries are represented in the numerical formulation using an immersed boundary algorithm capable of enforcing inviscid no-penetration boundary conditions as well as viscous skin friction with appropriate Reynolds number asymptotic properties using an equilibrium wall-model. The wall-model that provides a shear-stress constraint at the wall uses solution information interpolated via probing at a fixed distance of $1.5\Delta$ into the fluid away from walls, where $\Delta$ is the local grid spacing at the wall. Additional methods to damp generation of spurious numerical noise at walls as well as at coarse-fine interfaces are also utilized in the present work. All simulations presented in this work utilize the constant coefficient Vreman subgrid scale closure (21) to model the subgrid-scale stresses.

### 2.2.1 Computational Mesh

The Volcano ScaLES solver uses Cartesian grids generated using a recursive octree approach. Grids around engineering-quality configurations consist of an unstructured tree-of-cubes with millions of leaf cubes each containing a structured grid of $4^3$ or $8^3$ cells.

Figure 3 depicts the nested refinements regions for the optimized mesh. Refinement is emphasized for the boundary layer development from the nose of the car and for the wake and off-body vorticity directly behind the body. Section 3 of this paper discusses in detail the validation of this code against experimental data for the baseline windsor model and the rationale behind the choice of computational mesh.

### 2.3 Boundary Conditions

To follow the setup of the AutoCFD 3 and 4 workshops, the Windsor body is yawed 2.5 degrees and the Reynolds number is $Re = 2.9 \times 10^6$ based on the body length and the freestream velocity. A nominal inlet velocity of $40\,\mathrm{m\,s^{-1}}$ is given upstream of the car and a sensor is placed at $(x, y, z) = (-2\,\mathrm{m}, 1.3\,\mathrm{m}, 0\,\mathrm{m})$ to measure local static pressure and velocity. For the validation of the method against experimental data for the baseline geometry, these are used to normalize the simulation outputs, however for the dataset itself, the reference inlet velocity is used to normalize all outputs (see SI for further details),. The remaining boundary conditions are shown in Figure 2.

The domain required for this case (see Figure 2) represents the wind tunnel confinement but with the following modifications:

1. The Windsor body sidewalls are aligned with the $x$-direction, the tunnel is yawed.

2. Only the ground plane has a no slip condition and hence has boundary layer growth; the top and side walls should be treated as a slip or "inviscid" wall.

3. A long parallel inlet run is used in order to grow a boundary layer on the ground plane of approximately the correct thickness.

4. A parallel exit run is added downstream to avoid interactions with the wake. The domain extends upstream to $x = -5\,\mathrm{m}$ and downstream to $x = +6\,\mathrm{m}$ (the model is $x = -0.56\,\mathrm{m}$ nose to $x = +0.48\,\mathrm{m}$ base). The width and height of the CFD domain matches the wind tunnel.

5. The "outlet" for the tunnel is not a domain boundary for these simulations even though the volume data does not extend further. To minimize numerical reflections from the tunnel exit, the tunnel was allowed to vent into a large reservoir which is initialized with atmospheric conditions and given simple extrapolation boundary conditions far from the tunnel. This

means there is no fixed pressure condition or any other boundary condition enforced at the tunnel exit.

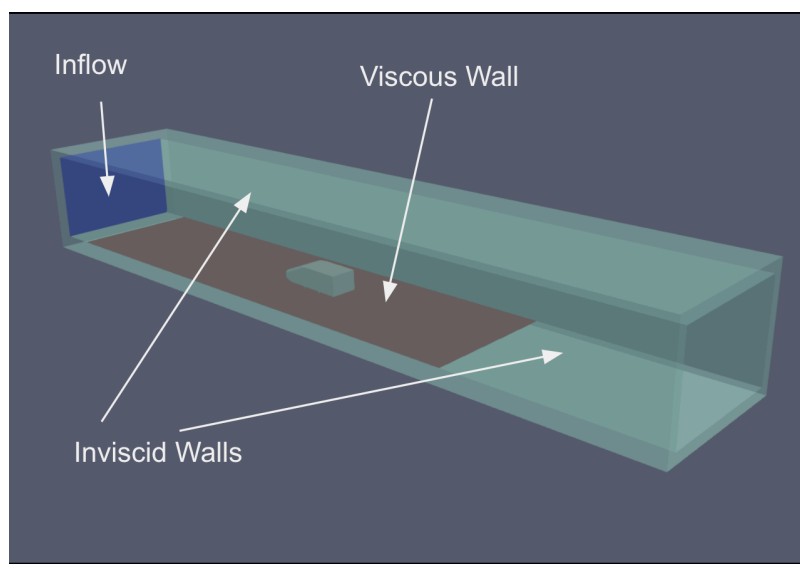

Figure 2: Depiction of the computational domain for the Windsor body simulations.

## 2.4 HPC setup

All simulations were run on Amazon Web Services, using a dynamic HPC cluster provisioned by AWS ParallelCluster v3.9. Amazon EC2 g5.48xlarge nodes were used for each case, which contain 8 Nvidia A10g GPU cards and a second generation AMD EPYC processor with 96 physical cores and 768GB of CPU memory. These are connected using a 100Gbit/s Elastic Fabric Adapter (EFA) interconnect (22). A 300TB Amazon Fsx for Lustre parallel file-system was used as the main data location during the runs, which were later transferred to object storage in Amazon S3. Each simulation was run on a single g5.48xlarge node i.e 8 A10g GPUs, and each case took approximately $2\,\mathrm{min}$ to mesh and $28\,\mathrm{h}$ to solve.

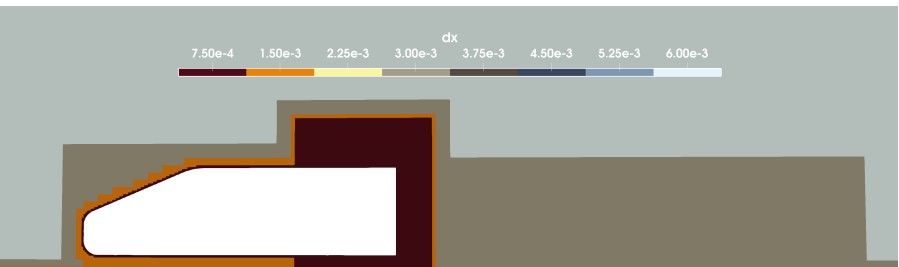

Figure 3: Grid spacing (dx) for a slice through $y = 0\,\mathrm{m}$ for the Windsor body.

Table 1: Computational cost for the Windsor body.

| Label | Min. grid spacing [mm] | No. of cells [$10^6$] | Wall time / CTU [min] | $C_d$ |
|---|---|---|---|---|
| $G_1$ | 1.00 | 151 | 6 | 0.31 |
| $G_2$ | 0.75 | 275 | 15 | 0.34 |
| $G_3$ | 0.625 | 468 | 27 | 0.33 |
| Exp. | - | - | - | 0.33 |

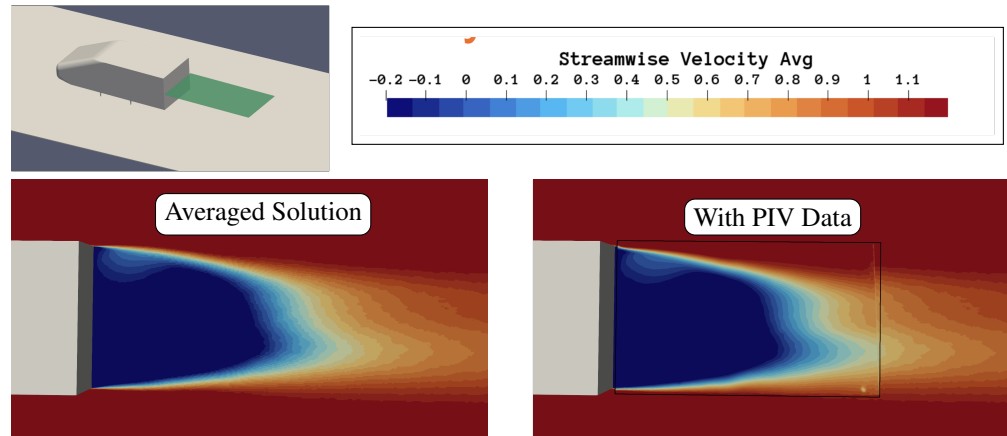

Figure 4: Time-averaged streamwise velocity at $y = 0.195\,\mathrm{m}$ with PIV data

## 3 Validation

For the design case study, we choose to keep the mesh refinement regions fixed for all design cases. Using the fixed refinement topology, we consider a consistent grid refinement study for the Windsor body. An example of a grid in this family is depicted in Figure 3. The surface of the body is refined to the finest level with a large wake block immediately behind the body. The block is taller than necessary for the Windsor body but since we are choosing to keep the refinement regions fixed we define a region to accommodate all designs ride height. Table 1 shows results of a consistent grid refinement study using a sequence of grids $\{G_j\}$ for $j = 1, 2$ and 3. The columns of the table show the minimum grid spacing used, the number of computational cells in the grid, the wall-clock time per convective time unit[4] (CTU) in minutes and the drag coefficient $C_d$. An additional row is included for the experimental drag coefficient value. Based on the results, $G_2$ is chosen as a representative baseline case and the remainder of the experimental comparisons are carried out for the $G_2$ grid. This was to achieve a balance between accuracy, computational cost and reach a mesh size that is typically used by the automotive industry for external aerodynamics.

**Velocity:** Typical of bluff-bodies, the Windsor body shows a large recirculation region at the base. The data for the $y = 0.195\,\mathrm{m}$ slice was taken with a freestream velocity of $40\,\mathrm{m\,s^{-1}}$ which is the same as our simulation but, in addition to this plane, tomographic PIV was taken for a 3D region behind the car. This data was recorded for a flow with a freestream velocity of $30\,\mathrm{m\,s^{-1}}$. To make comparisons with the tomographic PIV data, we normalize the velocities by their respective freestream velocity. With that caveat, Figures 4 & 5 shows a comparison of the normalized time-averaged streamwise velocity for $y = 0.195\,\mathrm{m}$ compared with the PIV measurements and the same solution at $z = 0\,\mathrm{m}$ compared to the tomographic PIV data. The overall shape of the wake region is well captured as well as is its interaction with the floor boundary layer which can be seen in the velocity contours. See SI for a detailed discussion on the validation of the underlying CFD method.

---

[4]Nondimensional time is defined to be $t^* = tU_\infty/L$ where $L$ is the length of the Windsor body, $U_\infty$ is the free-stream velocity, and $t$ is the time.

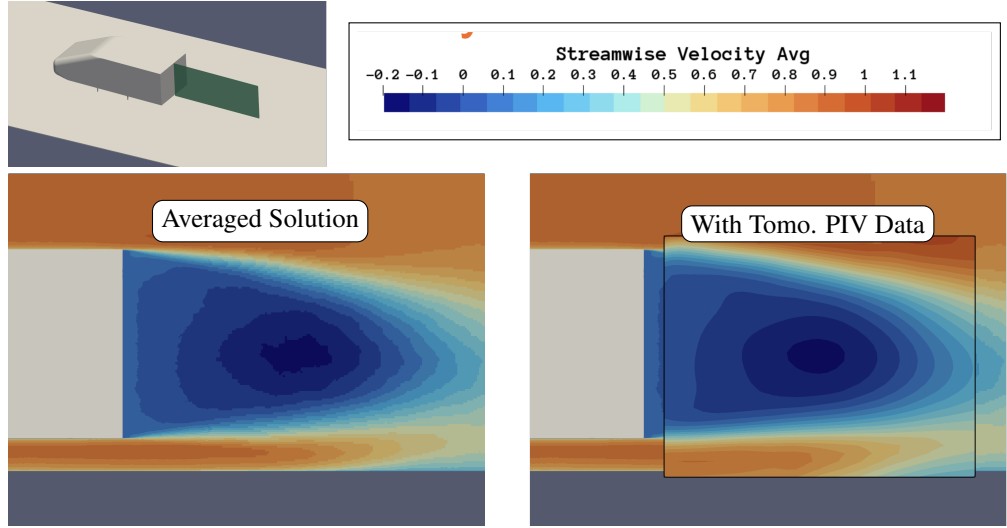

Figure 5: Time-averaged streamwise velocity at $z = 0\,\mathrm{m}$ with tomographic PIV data.

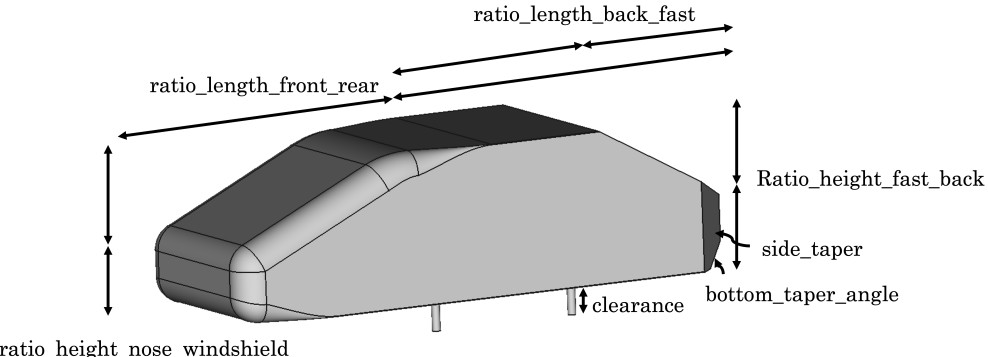

Figure 6: Parameterized CAD model

# 4 Dataset description

## 4.1 Geometry variations

The baseline geometry is transferred into a parameterized CAD model to allow for automated sampling in the design space. The chosen sampling algorithm is a Halton sequence, which results in a quasi-random low discrepancy sequence with the advantages of being deterministic and extendable. The parameters and respective ranges are shown in Figure 6 and Table 2. The parameters were chosen to provide a suitable range of geometries that would exhibit different flow patterns e.g pressure vs geometry induced separation. The min/max values of each value were based upon engineering judgement and to avoid completely unrealistic shapes as well as avoid invalid shapes during geometry creation. The choice of 355 geometries were also based upon a mixture of computational budget as well as matching what may be possible within industry i.e. how many geometries would realistically be generated within an engineering company. Future work could be to expand this dataset further if required.

The geometry choice within the dataset results in a broad range of flow physics - that is partly illustrated in Figures 7c and 7d which show the large range of drag and lift coefficients across the dataset. Figures 7a and 7b take two examples from the dataset that shows the resulting flow-field (mean streamwise velocity) for a geometry producing a high drag coefficient and another one for a low drag coefficient (see SI for more detailed discussion of the dataset outputs).

Table 2: Geometry variants of the Windsor body

| Variable | Min | Max |
|---|---|---|
| `ratio_length_front_rear` | 0 | 0.8 |
| `ratio_length_back_fast` | 0.08 | 0.5 |
| `ratio_height_nose_windshield` | 0.3 | 0.7 |
| `ratio_height_fast_back` | 0 | 0.9 |
| `side_taper [mm]` | 50 | 100 |
| `clearance [mm]` | 10 | 200 |
| `bottom_taper_angle [°]` | 1 | 50 |

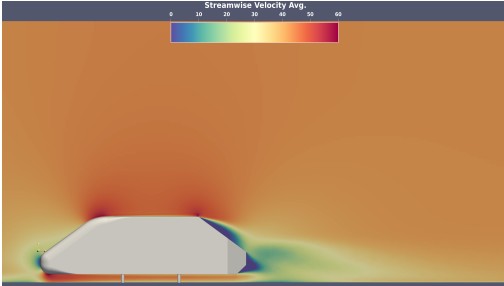

(a) Mean streamwise velocity for high drag geometry variant example

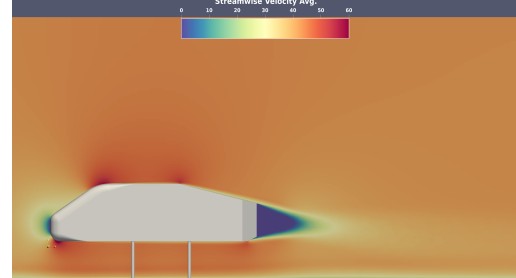

(b) Mean streamwise velocity for low drag geometry variant example

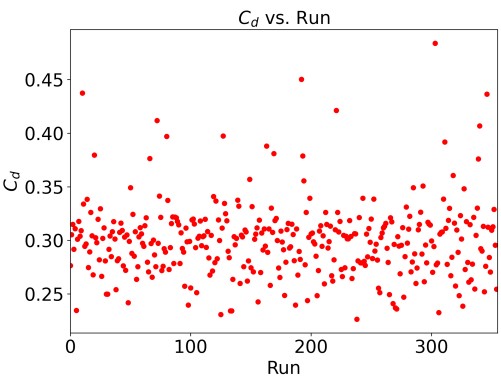

(c) Variation of drag coefficient against run number

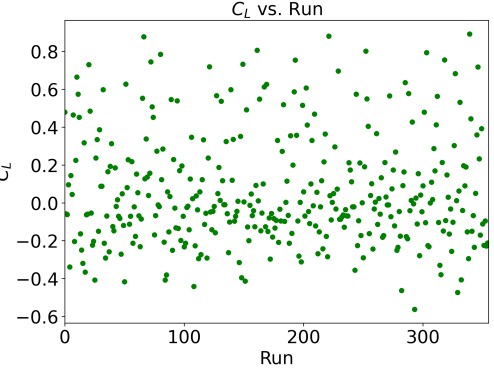

(d) Variation of lift coefficient against run number

Figure 7: Variation of mean streamwise velocity and force coefficients across a sample of the dataset

## 4.2 How to access the dataset

The dataset is open-source under the CC-BY-SA license [5] and available to download via Amazon S3. In order to download, no AWS account is required and the full details are provided in the SI, the dataset README [6] and the website [7]. Further mirroring sites are being explored and details will be on the website once available.

## 4.3 Dataset description

The dataset follows the same structure as two concurrently developed datasets; AhmedML (10) and DrivAerML (14). A summary of the dataset is provided in Table 3 (see SI for full details). The purpose of the dataset is to provide a rich dataset to support the development of a broad range of potential ML approaches. For this reason, we provide all possible outputs such as the full volume flow-field, the boundary surface, images of the flow-field as well as the time-averaged force and

---

[5]https://creativecommons.org/licenses/by-sa/4.0/)
[6]https://caemldatasets.s3.us-east-1.amazonaws.com/windsor/dataset/README.txt
[7]https://caemldatasets.org

moment coefficients. All the outputs are provided in commonly used open-source formats (vtp, vtu, etc.) which allow for the the broadest compatibility and ease of use for developers.

## 4.4 Limitations of the dataset

Whilst this high-fidelity, large-scale dataset has numerous benefits over prior work that was based upon lower-fidelity methods, there are a number of limitations:

- The dataset has purely geometrical differences with no variation in boundary conditions. Extending the dataset to include boundary condition changes, such as the inflow velocity, would help ML developers to use this dataset for more than just geometry prediction.
- Whilst the Windsor body improves over the Ahmed car body it lacks the complexity of a real-life vehicle (addressed by the upcoming related DrivAerML dataset (14))
- The dataset only includes time-averaged data rather than time-series. Future work could be to extend the dataset to also include a limited number of transient outputs i.e volume/boundary outputs at each time-step.

Table 3: Summary of the dataset contents

| Output | Description |
|---|---|
| Per run (inside each run_i folder) | |
| windsor_i.stl | surface mesh of the windsor body geometry |
| windsor_i.stp | surface CAD of the windsor body geometry |
| boundary_i.vtu | time-averaged flow quantities (Pressure Coefficient, Skin-Friction Coefficient, $y^+$) on the Windsor car body surface |
| volume_i.vtu | time-averaged flow quantities (pressure, velocity, Reynolds Stresses, Turbulent Kinetic Energy) within the domain volume |
| force_mom_i.csv | time-averaged drag, & lift, side-force and pitching moment coefficients using constant $A_{ref}$ & $L_{ref}$ |
| force_mom_varref_i.csv | time-averaged drag, & lift, side-force and pitching moment coefficients using case-dependant $A_{ref}$ |
| geo_parameters_i.csv | parameters that define the shape (in mm) |
| images | folder containing .png images of Reynolds stresses, streamwise velocity and pressure in $X, Y, Z$ slices through the volume |
| Other | |
| force_mom_all.csv | time-averaged drag & lift for all runs |
| force_mom_varref_all.csv | time-averaged drag & lift using case-dependant $A_{ref}$ for all runs |
| geo_parameters_all.csv | parameters that define the shape for all runs |

## ML evaluation

We conducted an example ML evaluation using a Graph Neural Network (GNN) approach, based upon a modified version of MeshGraphNets (23) (more details in the SI) to demonstrate how this dataset could be used to train a ML model to predict unseen cases. We assess two approaches; firstly directly predicting the Key Performance Indicators (KPIs); in this case the drag and lift coefficient values, for each geometry using the lower resolution STL surface mesh (89k nodes) as an input, and secondly predicting the lift and drag coefficient through integration of the surface wall-shear stress

and pressure from the higher resolution VTP surface mesh (4.4M nodes). Using the first method we find that using a 60/20/20 split of train, validation and test data, it is possible to obtain a MSE of less than 0.00028 for the drag coefficient and a MSE less than 0.0175 for the lift coefficient. An example of the prediction accuracy is shown in Figure 8 for the training, validation and test data for the drag coefficient. Using x8 Nvidia L40s GPUs (Amazon EC2 g6e.48xlarge instances via Amazon Web Services), the training completes in approximately 2hrs and the inference time for each new predicted geometry is 0.15 seconds. Additional results and further details of the ML setup are provided in the SI. Please note that these ML evaluations are preliminary and purely serve to illustrate how this dataset can be used for ML evaluation. We hope other groups will use this dataset to do a more thorough evaluation of different ML methodologies.

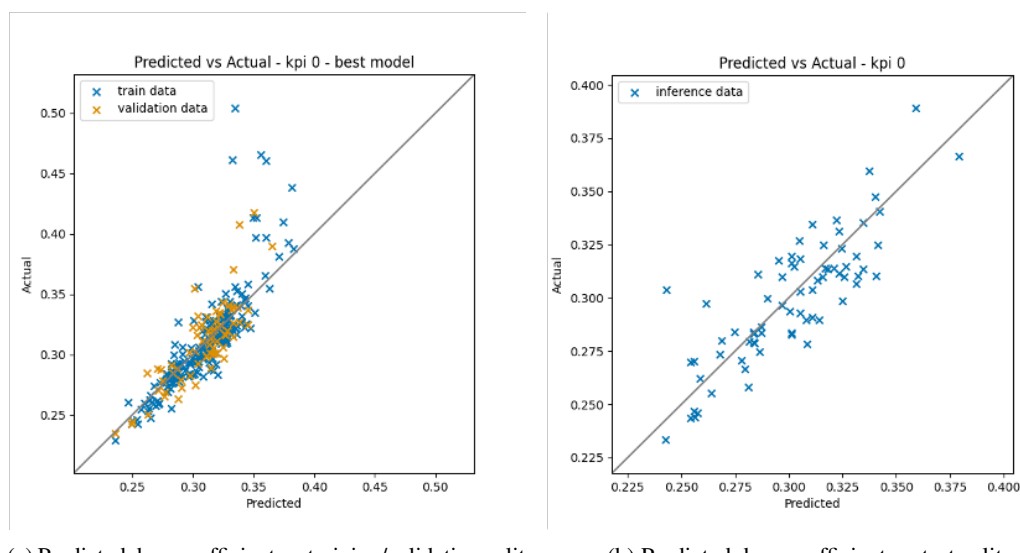

(a) Predicted drag coefficient on training/validation split    (b) Predicted drag coefficient on test split

Figure 8: Prediction of the drag coefficient using the direct KPI method

## 5  Conclusions

In this paper, we present a new large-scale, open-source Computational Fluid Dynamics dataset based upon 355 geometry variants of the automotive Windsor body. The flow conditions are fixed and seven geometric parameters of interest to automotive designers are varied. The CFD simulations use a high-fidelity, time-accurate scale-resolving approach to ensure accurate prediction of the underlying flow structures. The dataset uses widely used open-source formats and contains the vehicle geometries, time averaged integrated forces, time averaged vehicle surface data and time average volumetric flowfield data. It is hoped that this open-source dataset will allow for faster development of data-driven and physics-driven ML approaches for vehicle aerodynamics prediction.

## Acknowledgments and Disclosure of Funding

Thank you to members of the AutoCFD4 AI/ML TFG for early feedback on the dataset. Thank you to Nate Chadwick, Peter Yu, Mariano Lizarraga, Pablo Hermoso Moreno, Shreyas Subramanian and Vidyasagar Ananthan from Amazon Web Services for the useful contributions and feedback as well as debugging of the dataset in preparation for release. Also, thank you to Bernie Wang for providing additional feedback on the paper itself.

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
