# WindsorML: Supplementary Information

**Neil Ashton**[*]
Amazon Web Services
60 Holborn Viaduct
London, EC1A 2FD

**Jordan B. Angel**
Volcano Platforms Inc.
Stanford Research Park, 3240 Hillview Ave
Palo Alto, CA 94304

**Aditya S. Ghate**
Volcano Platforms Inc.
Stanford Research Park, 3240 Hillview Ave
Palo Alto, CA 94304

**Gaetan K. W. Kenway**
Volcano Platforms Inc.
Stanford Research Park, 3240 Hillview Ave
Palo Alto, CA 94304

**Man Long Wong**
Volcano Platforms Inc.
Stanford Research Park, 3240 Hillview Ave
Palo Alto, CA 94304

**Cetin Kiris**
Volcano Platforms Inc.
Stanford Research Park, 3240 Hillview Ave
Palo Alto, CA 94304

**Astrid Walle**
Siemens Energy
Huttenstraße 12, 10553
Berlin, Germany

**Danielle C. Maddix**
Amazon Web Services
410 Terry Ave. N.,
Seattle, WA 98109-5210, United States

**Gary Page**
Loughborough University
Epinal Way, Loughborough
LE11 3TU, United Kingdom

## Contents

[*]Now at NVIDIA, Corresponding Author: nashton@nvidia.com , contact@caemldatasets.org

38th Conference on Neural Information Processing Systems (NeurIPS 2024) Track on Datasets and Benchmarks.

# A  Numerical methodology

Volcano ScaLES, a GPU-based Large Eddy Simulation (LES) solver developed by Volcano Platforms Inc. is utilized to generate this database. Volcano ScaLES solves the compressible Navier-Stokes equations on Cartesian Octree grids with solid geometries discretized using an immersed boundary method suitable for high Reynolds number flows. In this section, we cover the basic formulation along with brief details about the closure modeling being employed.

## A.1  Large Eddy Simulations

Since a compressible Navier-Stokes formulation is utilized, Favre-averaged (density weighted) flow variables are solved

$$\tilde{f} = \frac{\overline{\rho f}}{\overline{\rho}} \tag{1}$$

where the $\bar{\cdot}$ denotes the standard LES low-pass filtering operator (with a length scale proportional to the grid spacing, $\Delta$), and $\rho$ denotes the density field. Using this combination of Favre-averaging and low-pass filtering, the conservation of mass equation can be written as

$$\frac{\partial \overline{\rho}}{\partial t} + \frac{\partial}{\partial x_i} \left( \overline{\rho} \tilde{u}_i \right) = 0 \tag{2}$$

where $\tilde{u}_i$ is the Favre-filtered velocity field. Similarly, the momentum equation is given by

$$\frac{\partial \overline{\rho} \tilde{u}_i}{\partial t} + \frac{\partial}{\partial x_j} \left( \overline{\rho} \tilde{u}_i \tilde{u}_j \right) + \frac{\partial \overline{p}}{\partial x_i} - \frac{\partial \hat{\sigma}_{ij}}{\partial x_j} = \frac{\partial R_{ij}}{\partial x_j} \tag{3}$$

where $\overline{p}$ is the low-pass filtered static pressure and the resolved viscous stress tensor is given by:

$$\hat{\sigma}_{ij} = 2\mu(\tilde{T}) \left( \frac{1}{2} \left( \frac{\partial \tilde{u}_i}{\partial x_j} + \frac{\partial \tilde{u}_j}{\partial x_i} \right) - \frac{1}{3} \delta_{ij} \frac{\partial \tilde{u}_k}{\partial x_k} \right) \tag{4}$$

Note that the dynamic viscosity, $\mu$ can be expressed as a function of Favre-filtered static temperature using an appropriate law (such as the Sutherland's law). However, in the present work, this dynamic viscosity has be assumed to a spatio-temporal constant with a value determined to achieve the required Reynolds number. Static temperature, $\tilde{T}$ is a function of the local pressure and density in accordance to the ideal gas law using a gas constant, $R$:

$$\tilde{T} = \frac{\overline{\rho T}}{\overline{\rho}} = \frac{\overline{P}}{R \overline{\rho}} \tag{5}$$

The term on the right-hand side of Equation 3 is the un-closed LES term that needs to be modelled,

$$R_{ij} = \overline{\rho} \left( \tilde{u}_i \tilde{u}_j - \widetilde{u_i u_j} \right) \tag{6}$$

Note that the residual stress given above assumes that $\hat{\sigma}_{ij} - \tilde{\sigma}_{ij}$ is negligible; this is true in high-Reynolds number flows and is exact for constant viscosity flows. Vreman's constant coefficient algebraic model(1) is used to close residual stress tensor:

$$R_{ij} = f_{\text{vreman}} \left( c_{\text{vreman}}, \Delta, \overline{\rho}, \frac{\partial \tilde{u}_i}{\partial x_j} \right) \tag{7}$$

The governing equation for conservation of total energy is given as

$$\frac{\partial \hat{e}}{\partial t} + \frac{\partial}{\partial x_j} \left( (\hat{e} + \overline{p}) \tilde{u}_j \right) + \frac{\partial \hat{q}_j}{\partial x_j} - \frac{\partial}{\partial x_j} \left( \hat{\sigma}_{ij} u_i \right) = R_e \tag{8}$$

where the unclosed terms on the right hand side represent interactions between subgrid-scale energy and the resolved scale energy. Since these terms are only relevant at high Mach numbers, the residual terms in the energy equation are modelled via just a turbulent Prandtl number and the Vreman's eddy viscosity model used for the momentum equations. The positive definiteness of the SGS eddy viscosity ensures that the numerical model for the filtered energy equation is purely dissipative, i.e. it assumes that the energy is exclusively transferred from the large scales to the small scales of motion.

$$R_e = f_{\text{vreman}}^e \left( c_{\text{vreman}}, \text{Pr}_{\text{turb}}, \Delta, \overline{\rho}, \frac{\partial \tilde{u}_i}{\partial x_j}, \frac{\partial \tilde{T}}{\partial x_j} \right) \tag{9}$$

The resolved total energy, $\hat{e}$ can be expressed exclusively in terms of other resolved field variables as

$$\hat{e} = \frac{\overline{p}}{\gamma - 1} + \frac{1}{2} \overline{\rho} \tilde{u}_i \tilde{u}_i \tag{10}$$

## A.2 Discretization

Volcano ScaLES solves the governing LES equations using a nominally 4th order spatially accurate finite difference discretization with favorable Kinetic Energy and Entropy Consistency properties making it suitable for Large Eddy Simulations (LES) over a vast range of flow regimes. The viscous flux discretization utilizes a mix of 4th order and 2nd order discretizations with high spectral bandwidth thereby allowing further robustness for high Reynolds number flows without additional numerical dissipation via the inviscid flux numerical operator in turbulence resolving regions. Physics-based numerical sensors that are functions of local velocity gradients as well as pressure and density fluctuations allow for spatio-temporally localized use of limiters needed to capture flows with discontinuities such as shocks and steep density gradients.

A key advantage of Cartesian grids with unit aspect ratio is the use of explicit time-stepping scheme even in flows involving highly complex geometries; the perfectly isotropic grid cells near the geometry do not introduce any numerical stiffness associated with complex grid topologies. There are other significant advantages of Cartesian meshes including significantly reduced memory requirements for storing the grid as well as efficient and high accuracy in the underlying numerical formulation. The popular Strong Stability Preserving (SSP) variant of the classical 3rd order Runge–Kutta scheme developed by Gottlieb & Shu(2) is utilized for all the work presented in this paper. Geometries are represented in the numerical formulation using a ghost-cell immersed boundary algorithm capable of enforcing inviscid no-penetration boundary conditions as well as viscous skin friction with appropriate Reynolds number asymptotic properties using an equilibrium wall-model. The wall-model that provides a shear-stress constraint at the wall uses solution information interpolated via probing at a fixed distance of $3/2\Delta$ into the fluid away from walls.

## B  Dataset

### B.1  Licensing terms

The dataset is provided with the Creative Commons CC-BY-SA v4.0 license[2]. A full description of the license terms is provided under the following URL:

```
https://caemldatasets.s3.us-east-1.amazonaws.com/windsor/dataset/LICENSE.
txt
```

### B.2  Access to dataset

The dataset is hosted on Amazon Web Services (AWS) via an Amazon S3 bucket

```
s3://caemldatasets/windsor/dataset
```

The dataset README.txt will be kept up to date for any changes to the dataset and can be found at the following URL:

```
https://caemldatasets.s3.us-east-1.amazonaws.com/windsor/dataset/README.txt
```

The dataset itself can be downloaded via the AWS Command Line Interface (CLI) tool, which is free-of-charge. An instruction about how to install the AWS CLI tool is given here: `https://docs.aws.amazon.com/cli/latest/userguide/getting-started-install.html`. After installing AWS CLI, please follow the example instructions in the README.txt on ways to download the data from Amazon S3 e.g to download all the files you can run the following (however note the dataset size is more than 20TB in total) :

```
  aws s3 cp --recursive s3://caemldatasets/windsor/dataset .
```

Note 1 : If you don't have an AWS account you will need to add –no-sign-request within your AWS command i.e aws s3 cp –no-sign-request –recursive etc...
Note 2 : If you have an AWS account, please note the bucket is in us-east-1, so you will have the fastest download if you have your AWS service or EC2 instance running in us-east-1.

### B.3  Long-term hosting/maintenance plan

The data is hosted on Amazon S3 as it provides high durability, fast connectivity and is accessible without first requesting an account or credentials (only AWS CLI tools, described above, which are free to download and use). A dedicated website has been created for this dataset and the two associated datasets; AhmedML (3) and DrivAerML datasets (4) at https://caemldatasets.org to help further clarify where the data is hosted and to communicate any additional mirroring sites.

### B.4  Intended use

The dataset was created with the following intended uses:

- Development and testing of data-driven or physics-driven ML surrogate models for the prediction of surface, volume and/or force coefficients
- a stepping stone dataset between the simpler AhmedML (3) and more complex DrivAerML dataset (4)
- Large-scale dataset to study bluff-body flow physics

### B.5  DOI

At present there is no specific DOI for the dataset (only for this associated paper) - however the authors are investigating ways of assigning DOI to this Amazon S3 hosted dataset.

---

[2]`https://creativecommons.org/licenses/by-sa/4.0/deed.en`

## B.6  Dataset contents

For each geometry there is a separate folder (e.g `run_1`, `run_2`, ..., `run_i`, etc.) , where "i" is the run number that ranges from 0 to 354. All run folders contain the same time-averaged data which is summarized in Table 1.

Table 1: Summary of the dataset contents

| Output | Description |
|---|---|
| Per run (inside each run_i folder) | |
| windsor_i.stl | surface mesh of the windsor body geometry |
| windsor_i.stp | surface CAD of the windsor body geometry |
| boundary_i.vtu | time-averaged flow quantities (Pressure Coefficient, Skin-Friction Coefficient, $y^+$) on the Windsor car body surface |
| volume_i.vtu | time-averaged flow quantities (pressure, velocity, Reynolds Stresses, Turbulent Kinetic Energy) within the domain volume |
| force_mom_i.csv | time-averaged drag, & lift, side-force and pitching moment coefficients using constant $A_{ref}$ & $L_{ref}$ |
| force_mom_varref_i.csv | time-averaged drag, & lift, side-force and pitching moment coefficients using case-dependant $A_{ref}$ |
| geo_parameters_i.csv | parameters that define the shape (in mm) |
| images | folder containing .png images of Reynolds stresses, streamwise velocity and pressure in $X, Y, Z$ slices through the volume |
| Other | |
| force_mom_all.csv | time-averaged drag & lift for all runs |
| force_mom_varref_all.csv | time-averaged drag & lift using case-dependant $A_{ref}$ for all runs |
| geo_parameters_all.csv | parameters that define the shape for all runs |

### B.6.1  Cartesian to Unstructured output

In order to simplify the reading of the dataset into existing tools, the Cartesian data was converted into a fully unstructured VTK mesh made of hexahedra. Solution data at cell centers of the hexahedra are averaged to cell corners accounting for block neighbor's data. This change makes the data easier to consume since VTK is widely used in the scientific community and tools to read and process these files are readily available (e.g. ParaView).

### B.6.2  Time-Averaging

The dataset only includes time-averaged data rather than time-series due to the extremely large number of time-steps that would be required to output i.e $> 100,000$. However future work could be to include this for a limited number of runs to help develop models to capture the time-history.

### B.6.3  Force Coefficients

The drag, lift and side-force coefficients are defined as follows (please note that for all the simulations in this work $y$ is upwards, which differs to the original Windsor work where $z$ is upwards):

$$C_{\mathrm{D}} = \frac{F_x}{0.5\,\rho_\infty\,|U_\infty|^2\,A_{ref}}\,, \quad C_{\mathrm{L}} = \frac{F_y}{0.5\,\rho_\infty\,|U_\infty|^2\,A_{ref}}\,, \quad C_{\mathrm{S}} = \frac{F_z}{0.5\,\rho_\infty\,|U_\infty|^2\,A_{ref}}\,, \quad (11)$$

Where $F$ is the integrated force, $A$ is the frontal area of the geometry, and $\rho_\infty$ is the reference density (see Table 2 for reference values).

The pitching moment is defined as:

$$C_{\mathrm{My}} = \frac{M_y}{0.5\,\rho_\infty\,|U_\infty|^2\,A_{ref}\,L_{ref}}\,, \tag{12}$$

Where $M$ is the integrated moment, $A_{ref}$ is the frontal area of the geometry, $L_{ref}$ is the reference length, and $\rho_\infty$ is the reference density (see Table 2 for reference values).

Two outputs are provided for the forces; one in which the reference area is kept constant across all details (force_mom_i.csv), and secondly one where it based upon the frontal-area of each different geometry variant (force_mom_varref_i.csv).

### B.6.4 Boundary data

Time-averaged surface data for nondimensional force coefficients are provided. The surface pressure coefficient $C_p = (p - p_{\mathrm{ref}})/q_{\mathrm{ref}}$ where the "freestream" dynamic pressure is $q_{\mathrm{ref}} = \frac{1}{2}\rho_{\mathrm{ref}}|U_{\mathrm{ref}}|^2$ and $p_{\mathrm{ref}}$ is the "freestream" pressure. Similarly, $cf_x$, $cf_y$ and $cf_z$ correspond to the wall-strear-stress tensor (units Pascal) projections in the three coordinate directions normalized by $q_{\mathrm{ref}}$. Lift and drag coefficients can be computed via surface integration of the local $C_p$, $cf_x$, etc. using a simple surface quadrature rule (rectangle-rule in the present case). Note that lift and drag use the "reference area" to normalize the differential area that shows up for surface integration.

As previously mentioned, the time-averaged quantities from the numerical sensor (measured at $(x, y, z) = (-2\,\mathrm{m}, 1.3\,\mathrm{m}, 0\,\mathrm{m})$) are used to normalize the force and pressure coefficients for the baseline experimental setup, but not for this particular dataset (given there is no wind-tunnel data for the geometries).

Table 2: Nominal freestream conditions and reference dimensions.

| | |
|---|---|
| $U_{\mathrm{ref}}$ | 40 m/s |
| $\rho_{\mathrm{ref}}$ | 1.31 kg/m$^3$ |
| $p_{\mathrm{ref}}$ | 37330.4 Pa |
| $A_{\mathrm{ref}}$ | 0.112 m$^2$ |
| $L_{\mathrm{ref}}$ | 0.6375 m |

### B.6.5 Volume data

The "outlet" for the tunnel is not a domain boundary for these simulations even though the volume data does not extend further. To minimize numerical reflections from the tunnel exit, the tunnel was allowed to vent into a large reservoir which is initialized with atmospheric conditions and given simple extrapolation boundary conditions far from the tunnel. This means there is no fixed pressure condition or any other boundary condition enforced at the tunnel exit.

### B.6.6 Images

Table 4 describes the .png images that are created in $x$, $y$ and $z$ directions to either enable ML methods to directly predict them and/or provide a quick way to inspect the flow fields.

### B.6.7 Data formats

All provided data is in the open source format VTK (i.e. *.vtp and *.vtu) to ensure the broadest compatibility.

Table 3: List of output quantities in the provided dataset files, all quantities are time-averaged.

| Symbol | Units | Field name | Description |
|---|---|---|---|
| **volume_i.vtu** | | | |
| $\overline{p^*}$ | $[\mathrm{m}^2/\mathrm{s}^2]$ | pressureavg | relative kinematic pressure |
| $\overline{U_x}$ | $[\mathrm{m/s}]$ | velocityxavg | velocity component in $x$ |
| $\overline{U_y}$ | $[\mathrm{m/s}]$ | velocityyavg | velocity component in $y$ |
| $\overline{U_z}$ | $[\mathrm{m/s}]$ | velocityzavg | velocity component in $z$ |
| $\overline{u'_x u'_x}$ | $[\mathrm{m}^2/\mathrm{s}^2]$ | reynoldsstressxx | resolved Reynolds stress xx |
| $\overline{u'_y u'_y}$ | $[\mathrm{m}^2/\mathrm{s}^2]$ | reynoldsstressyy | resolved Reynolds stress yy |
| $\overline{u'_z u'_z}$ | $[\mathrm{m}^2/\mathrm{s}^2]$ | reynoldsstresszz | resolved Reynolds stress zz |
| $\overline{u'_x u'_y}$ | $[\mathrm{m}^2/\mathrm{s}^2]$ | reynoldsstressxy | resolved Reynolds stress xy |
| $\overline{u'_x u'_z}$ | $[\mathrm{m}^2/\mathrm{s}^2]$ | reynoldsstressxz | resolved Reynolds stress xz |
| $\overline{u'_y u'_z}$ | $[\mathrm{m}^2/\mathrm{s}^2]$ | reynoldsstressyz | resolved Reynolds stress yz |
| $\overline{k}$ | $[\mathrm{m}^2/\mathrm{s}^3]$ | tke | turbulent kinetic energy |

| Symbol | Unit | Field name | Description |
|---|---|---|---|
| **boundary_i.vtu** | | | |
| $\overline{y^+}$ | $[-]$ | yplusavg | $y^+$ |
| $\overline{C_f x}$ | $[-]$ | cfxavg | skin-friction coefficient in $x$ |
| $\overline{C_f y}$ | $[-]$ | cfyavg | skin-friction coefficient in $y$ |
| $\overline{C_f z}$ | $[-]$ | cfzavg | skin-friction coefficient in $z$ |
| $\overline{C_p}$ | $[-]$ | cpavg | static pressure coefficient |
| $\overline{C_p}^2$ | $[-]$ | cpvar | static pressure variance coefficient |

Table 4: Description of files within the images folder

| Name | Description |
|---|---|
| pressureavg (folder) | time-averaged pressure |
| velocityxavg (folder) | time-averaged $x$-component of the velocity |
| rstress_xx (folder) | time-averaged $xx$-component of the Reynolds stress tensor |
| rstress_yy (folder) | time-averaged $yy$-component of the Reynolds stress tensor |
| rstress_zz (folder) | time-averaged $zz$-component of the Reynolds stress tensor |
| windsor_i.png | picture of the geometry itself |
| view1_constz_scan_0000-0009.png | 10 slices of the above variables in z from z= -0.4 to 0.4 |
| view2_constx_scan_0000-0079.png | 80 slices of the above variables in x from x=-0.5 to 1.0 |
| view3_consty_scan_0000-0019.png | 20 slices of the above variables in y from y=0.03 to 0.55 |

## B.7 Geometry variants

The baseline geometry is transferred into a parameterized CAD model to allow for automated sampling in the design space. The chosen sampling algorithm is a Halton sequence, which results in a quasi-random low discrepancy sequence with the advantages of being deterministic and extendable. The parameters and respective ranges are shown in Figure 1 and Table 5. The parameters were chosen to provide a suitable range of geometries that would exhibit different flow patterns e.g pressure vs geometry induced separation. The min/max values of each value were based upon engineering judgement and to avoid completely unrealistic shapes as well as avoid invalid shapes during geometry creation. The choice of 355 geometries were also based upon a mixture of computational budget as well as matching what may be possible within industry i.e how many geometries would realistically be generated within an engineering company. Future work could be to expand this dataset further if required.

The geometry choice within the dataset results in a broad range of flow physics - that is partly illustrated in Figures 4 and 5 which show the large range of drag and lift coefficients across the dataset. Figures 2 and 3 take two examples from the dataset that shows the resulting flow-field (mean streamwise velocity) for a geometry producing a high drag coefficient and another one for a low drag coefficient.

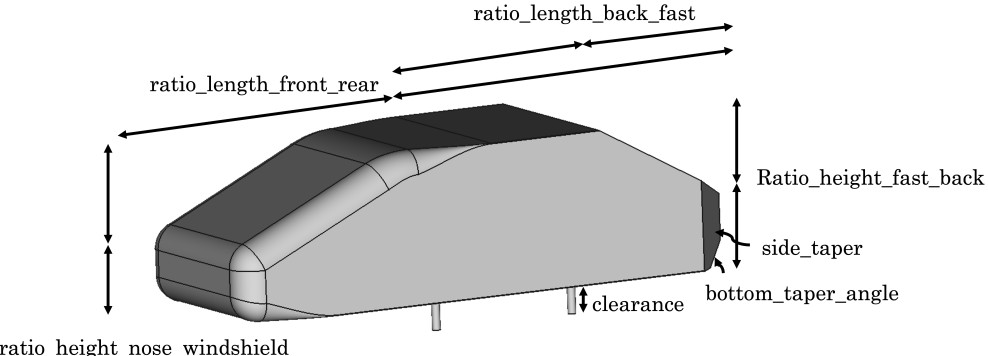

Figure 1: Parameterized CAD model

Table 5: Geometry variants of the Windsor body

| Variable | Min | Max |
|---|---|---|
| ratio_length_front_rear | 0 | 0.8 |
| ratio_length_back_fast | 0.08 | 0.5 |
| ratio_height_nose_windshield | 0.3 | 0.7 |
| ratio_height_fast_back | 0 | 0.9 |
| side_taper [mm] | 50 | 100 |
| clearance [mm] | 10 | 200 |
| bottom_taper_angle [°] | 1 | 50 |

Figures 6, 7, 8 & 9 are provided to give an illustration of the range of flow features and post-processing that makes this a rich dataset for the development and testing of ML methods.

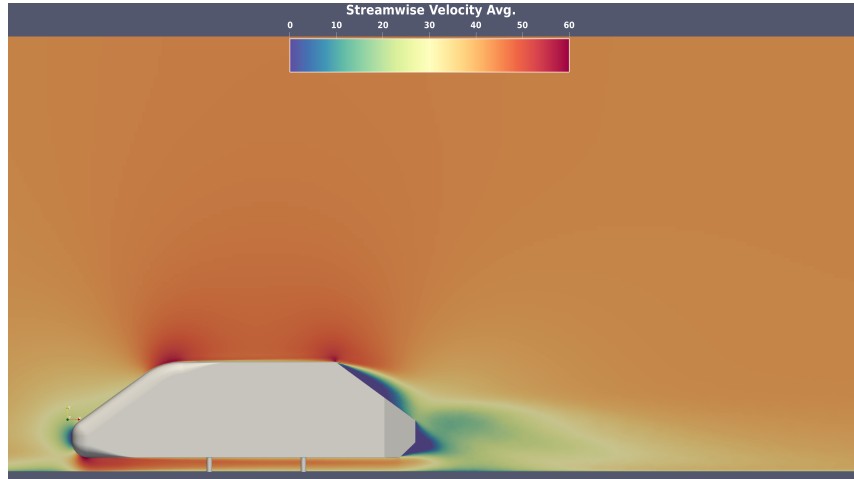

Figure 2: Mean streamwise velocity for high drag geometry variant example

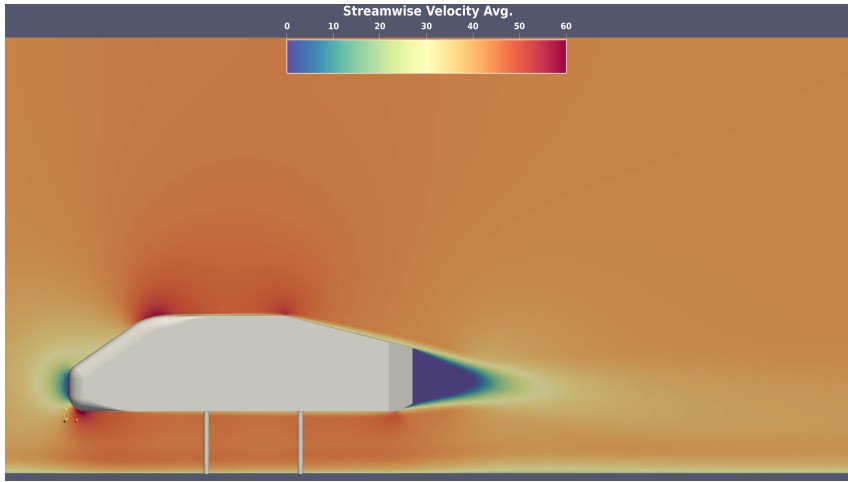

Figure 3: Mean streamwise velocity for low drag geometry variant example

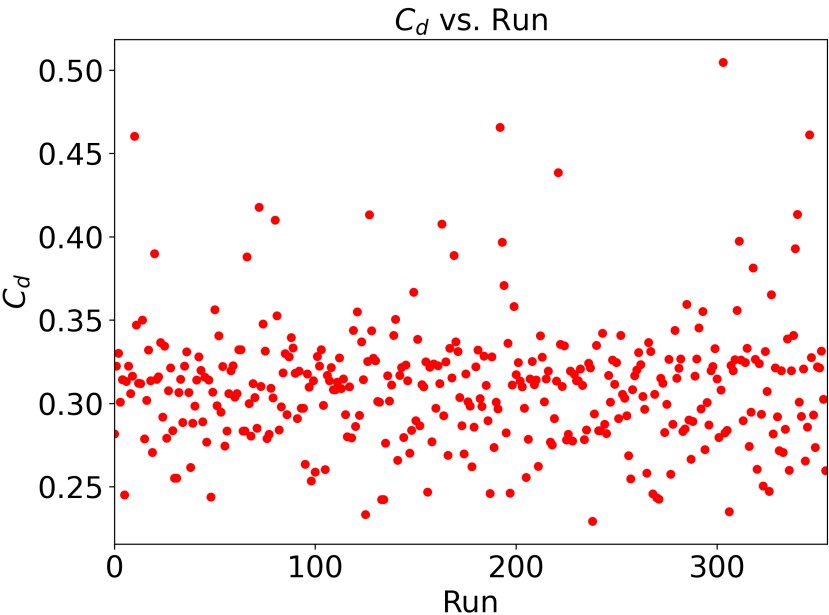

Figure 4: Variation of drag coefficient against run number

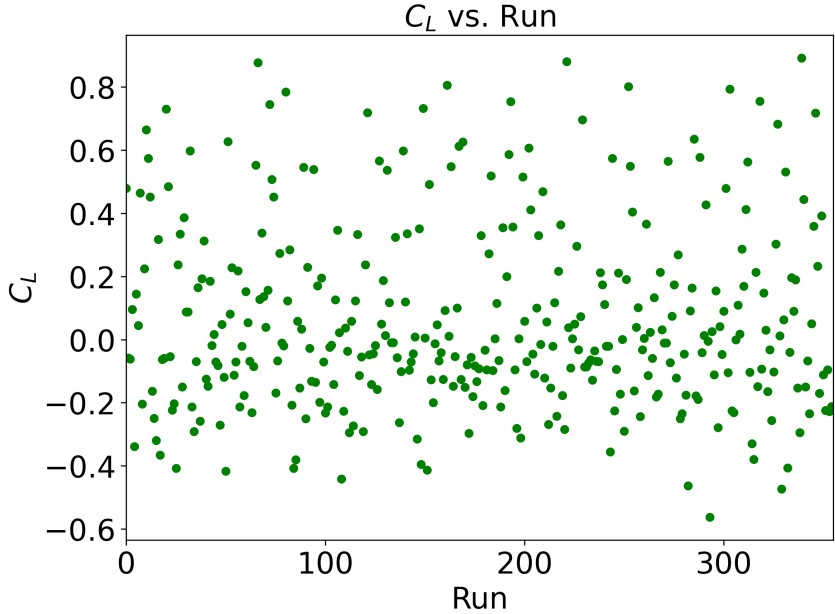

Figure 5: Variation of lift coefficient against run number

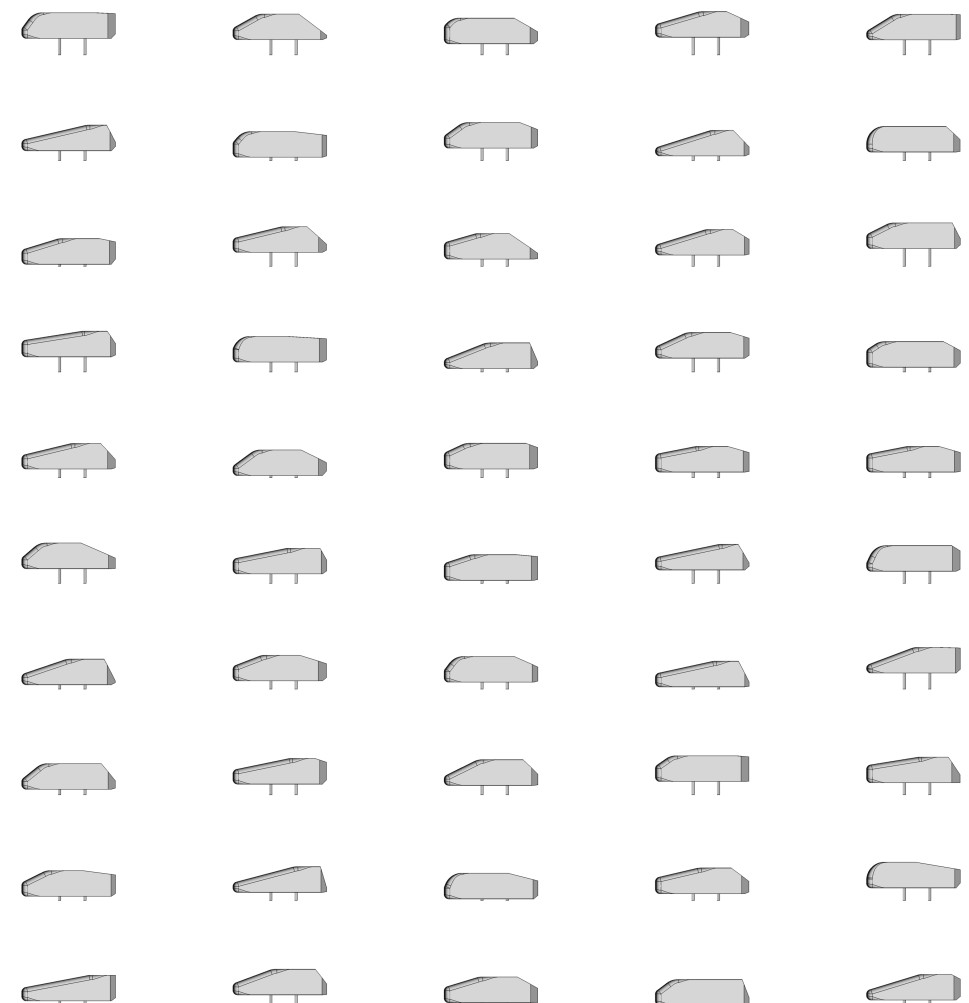

Figure 6: Image of the geometry for runs 1 to 50

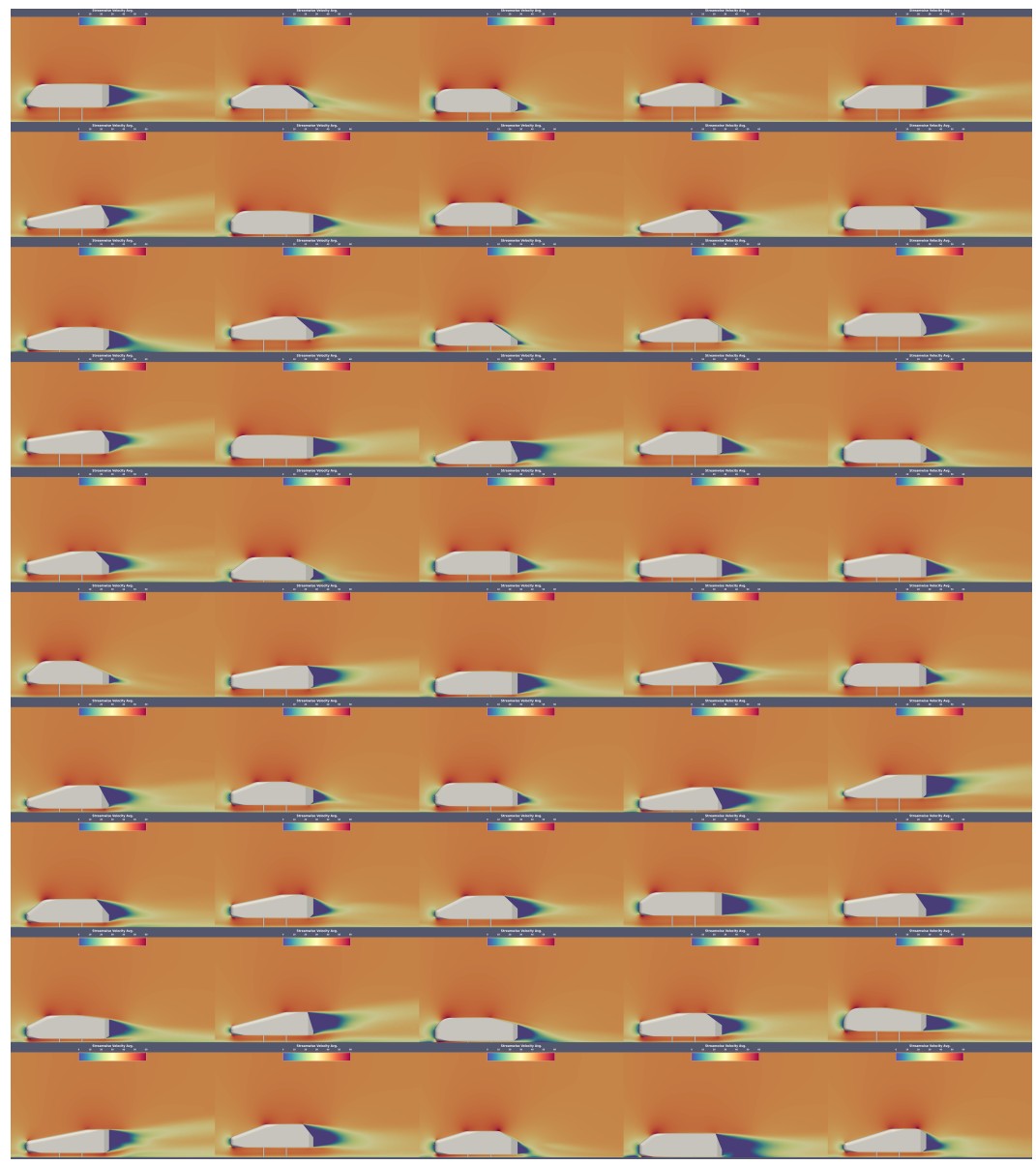

Figure 7: Mean Streamwise velocity for runs 1 to 50 at $z = 0$

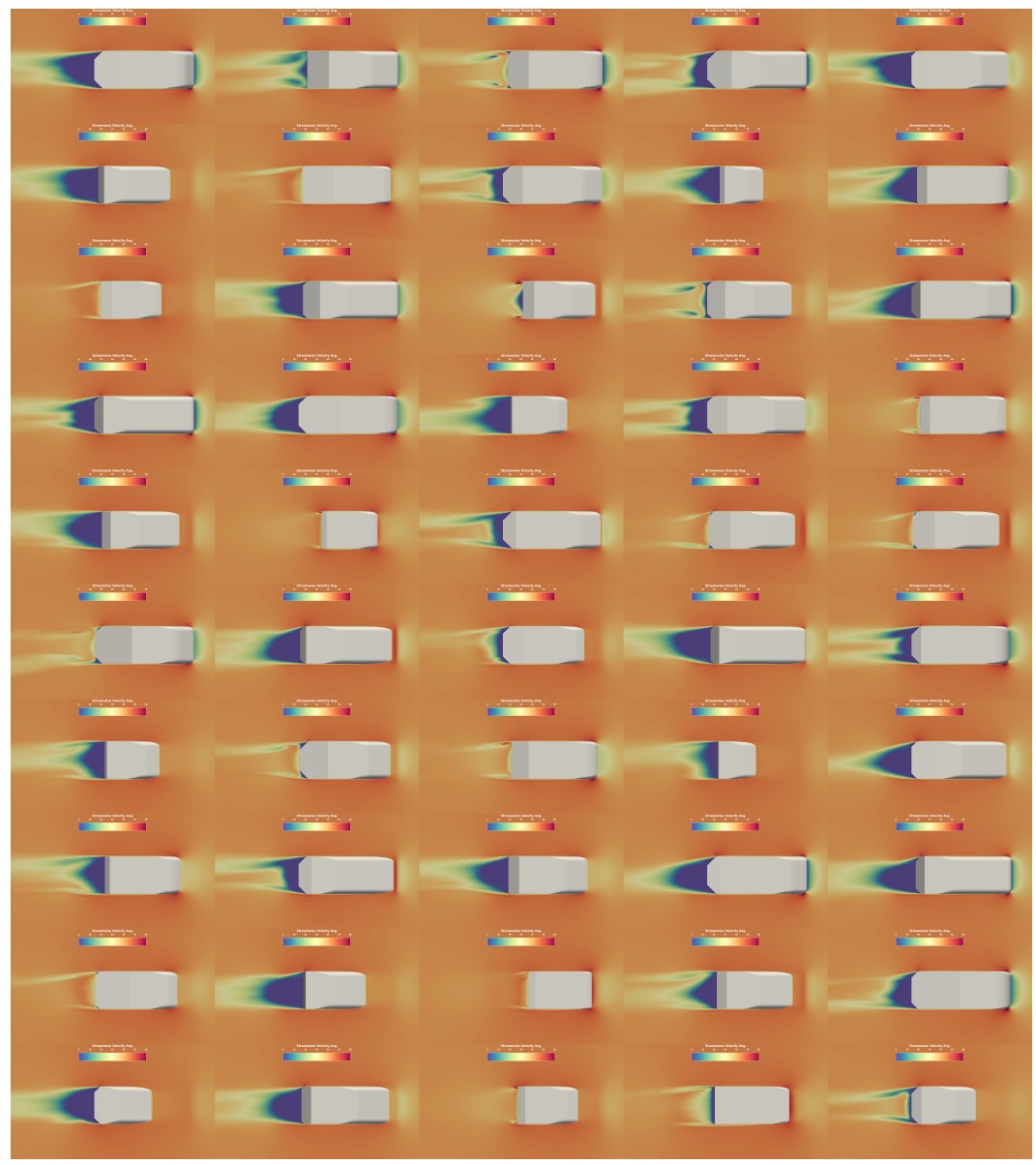

Figure 8: Mean Streamwise velocity for runs 1 to 50 at $y = 0$

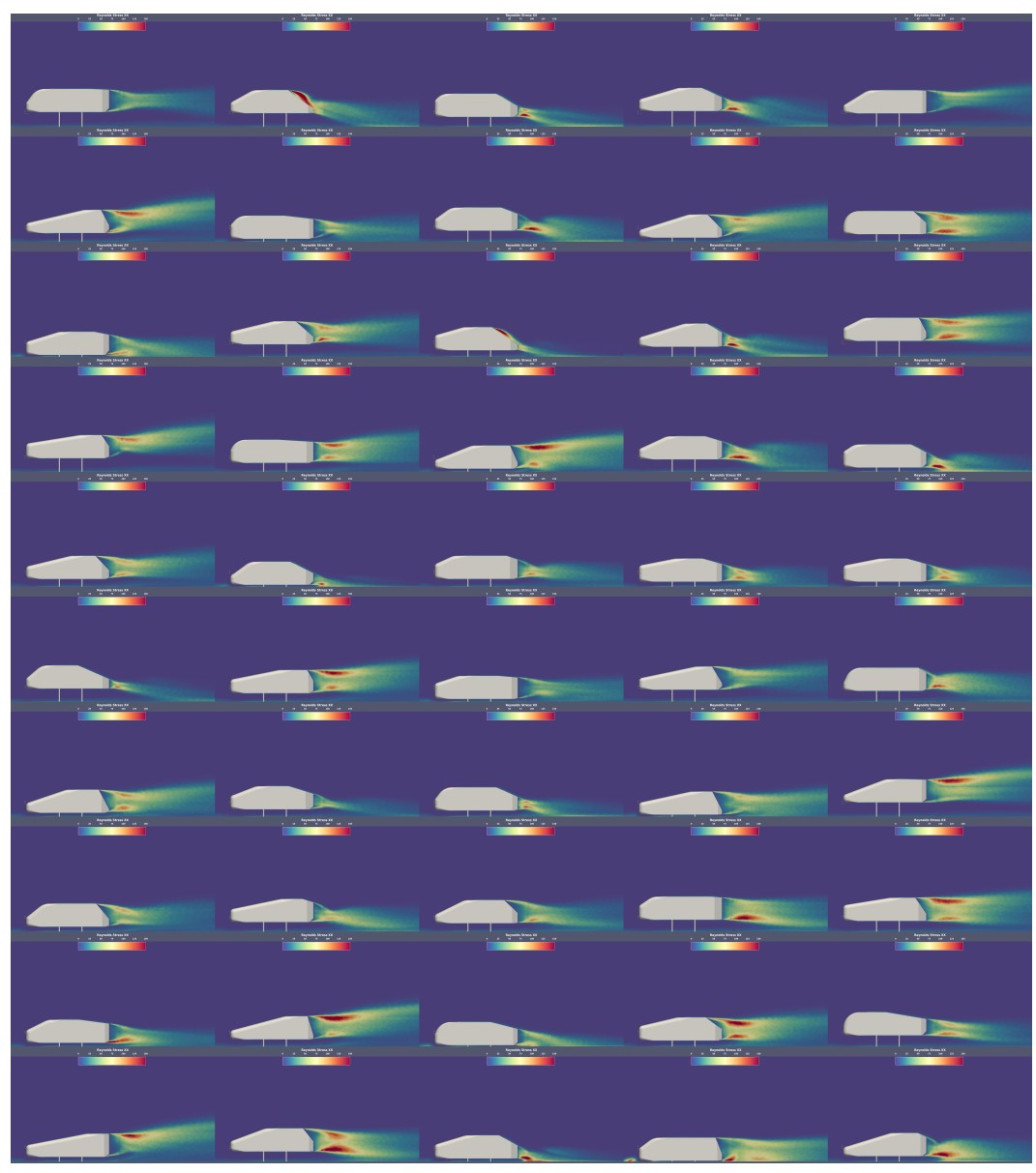

Figure 9: Mean Reynolds Stress (xx) for runs 1 to 50 at $z = 0$

# C  Additional Validation details

This sections provide some additional validation details for the Windsor body case left out of the main paper. The main paper showed grid convergence results for the drag coefficient which motivated our choice of the $G_2$ mesh as our baseline grid. Here we consider local experimental data in the form of surface pressure measurements and velocity PIV data.

We consider an additional modification to the problem to improve efficiency. The Mach number of the problem is not actually specified but is implied for some temperature that must be assumed. This degree of freedom can be chosen to artificially increase the Mach number while keeping the Reynolds number fixed. Assuming standard atmospheric conditions implies a freestream Mach number $M \approx 0.11$. By instead taking the ambient temperature to be $T \approx 100\,\mathrm{K}$, the Mach number increases to $M = 0.2$ which increases the maximum stable timestep by $55\%$. Figure 16 shows the pressure coefficient comparisons for $y = 0.2595\,\mathrm{m}$ and $z = 0\,\mathrm{m}$ with the original Mach number, the increased Mach number case and the experimental data. Increasing the Mach number did not degrade the quality of the solution but significantly reduces the turn around time for each simulation considered in the dataset. Comparisons of base pressure and PIV data are shown below for the $G_2$ grid at $M = 0.2$. As defined in the AutoCFD3 Case 1 problem description, a numerical sensor is placed in the tunnel and average flow quantities from this probe are used to normalize the pressure and velocities from the simulation.

**Pressure:** In Figure 10 and Figure 11, we show experimental and computed surface pressure coefficient values, $C_p$, for the cut planes $z = 0\,\mathrm{m}$ and $y = 0.2595\,\mathrm{m}$ respectively. For the $z = 0\,\mathrm{m}$ plane we observe stronger suction in the CFD compared to the experiment near $x = -0.5\,\mathrm{m}$ where the flow transitions to turbulence, which is typical in other CFD simulations for this problem. Otherwise, there is good agreement for the upper and lower sections. For the $y = 0.2595\,\mathrm{m}$ cut, the CFD again shows a stronger suction peak this time in the A-pillar area. Small variations in the data suggest a longer averaging could be done but it would not be expected to change the agreement. Overall, the agreement is reasonably good and similar to other results seen in the AutoCFD3 results. A comparative view of the base pressure is shown in Figure 12 where similar pressure distributions are seen with a low pressure region on the leeward side of the vehicle associated with the recirculation region. The simulation results show a higher pressure than the experiment which is most evident on the $-z$ side, however, the spatial distribution of the pressure and thus the shape of the contours match well.

**Velocity:** Time-averaged streamlines colored by normalized mean streamwise velocity at $y = 0.195\,\mathrm{m}$ are shown in Figure 13 compared with experimental data from the thesis by Varney (5). The streamlines show the large recirculation region behind the vehicle and the correct flow topology. The data for the $y = 0.195\,\mathrm{m}$ slice was taken with a freestream velocity of $40\,\mathrm{m\,s}^{-1}$ which is the same as our nominal freestream velocity but, in addition to this plane, tomographic PIV was taken for a 3D region behind the car. This data was recorded for a flow with a freestream velocity of $30\,\mathrm{m\,s}^{-1}$. To make comparisons with the tomographic PIV data, we normalize their velocities by their freestream velocity. Figures 14 and 15 show progressive cuts through the wake going downstream. Starting nearest the car, the experiment shows a thicker shear layer but as we move downstream the predicted and experimental flow have essentially the same shear layer thickness. The overall shape of the wake region is well captured as well as its interaction with the floor boundary layer which can be seen in the velocity contours.

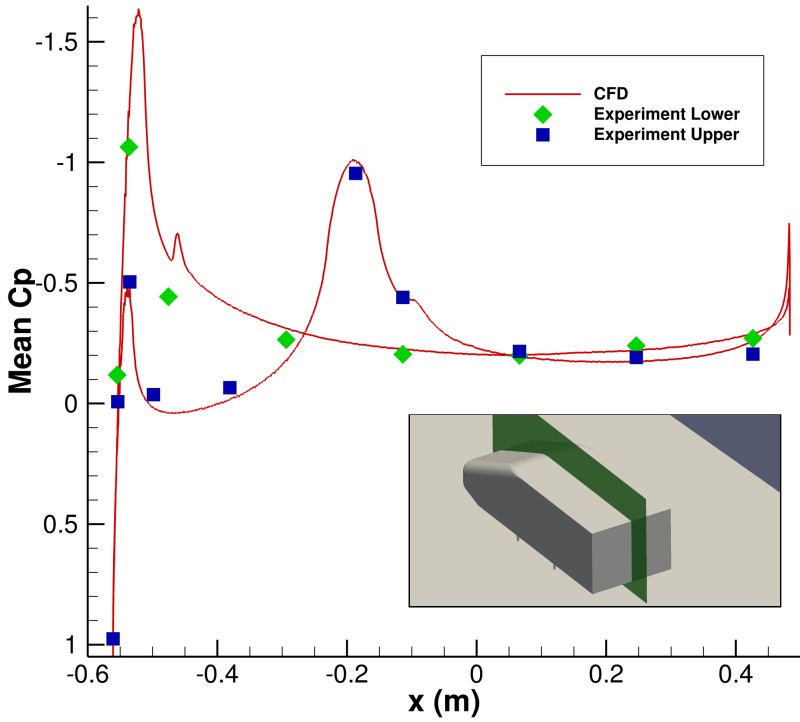

Figure 10: Comparison of mean pressure coefficient at $z = 0\,\mathrm{m}$ for the Windsor body.

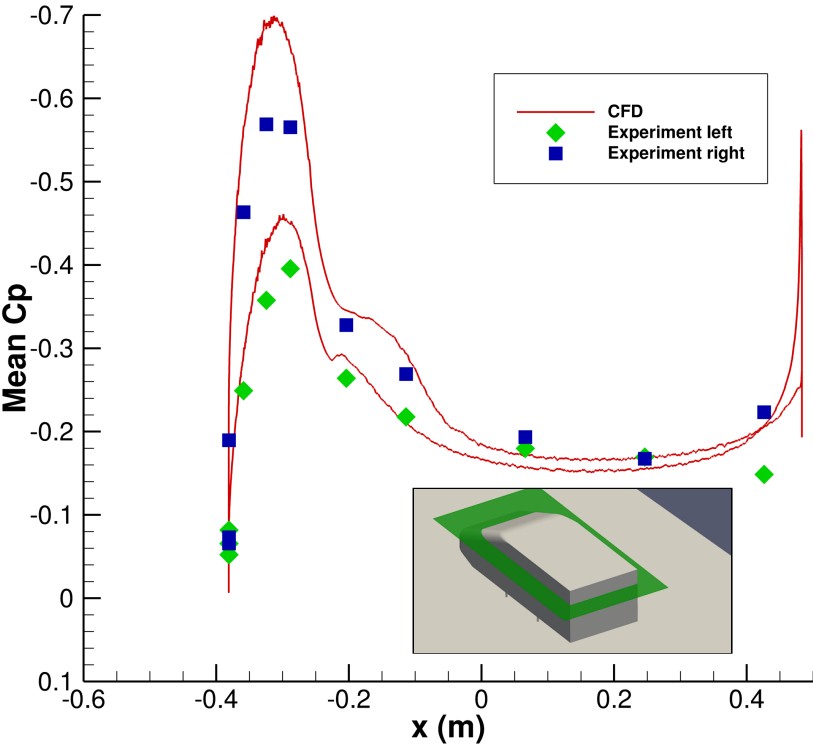

Figure 11: Comparison of pressure coefficient at $y = 0.2595\,\mathrm{m}$ for the Windsor body.

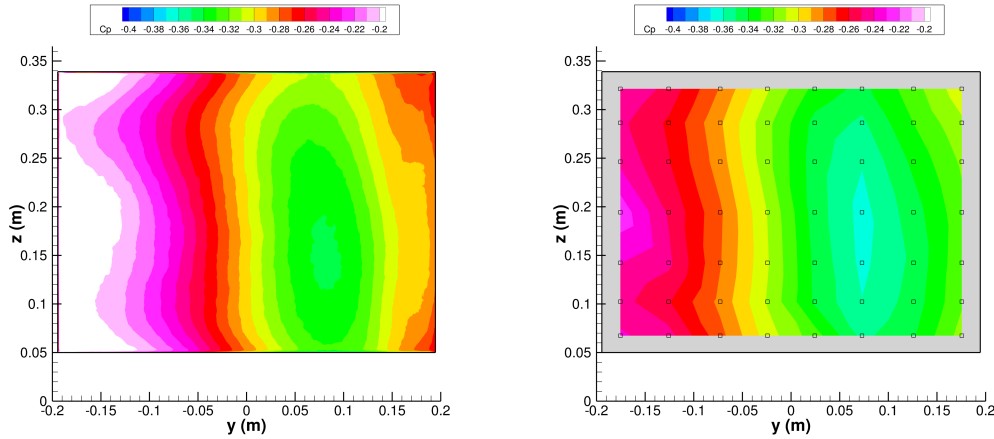

Figure 12: Left: Time-averaged pressure coefficient at the base of the Windsor body. Right: Comparison of time-averaged pressure coefficient at the base of the Windsor body.

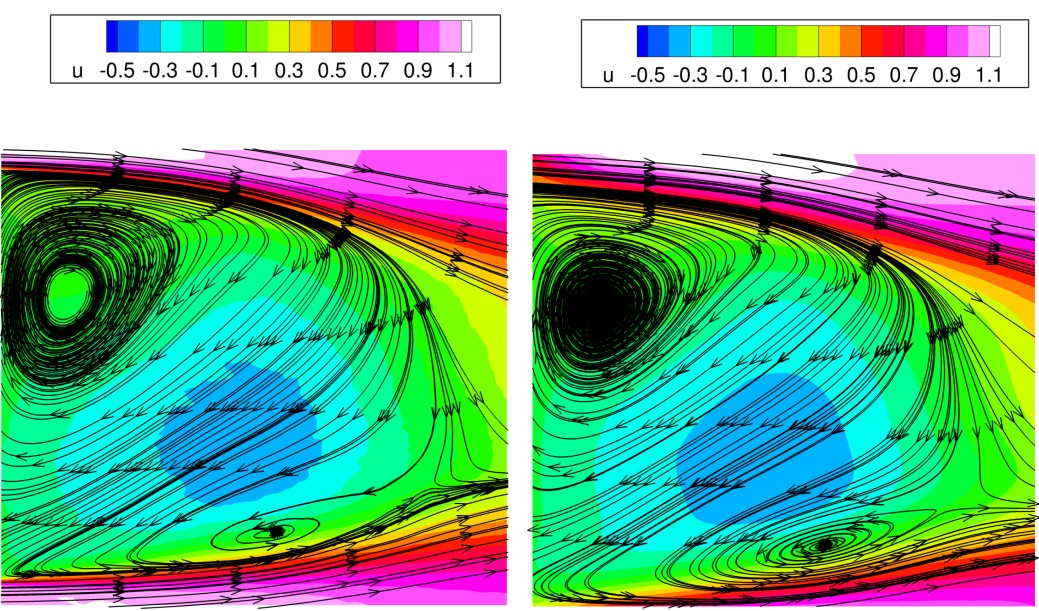

Figure 13: Comparison of mean velocity streamlines colored by normalized streamwise velocity at $y = 0.195\,\mathrm{m}$ for the Windsor body. CFD results are shown on the left and the experiment data is plotted on the right.

.

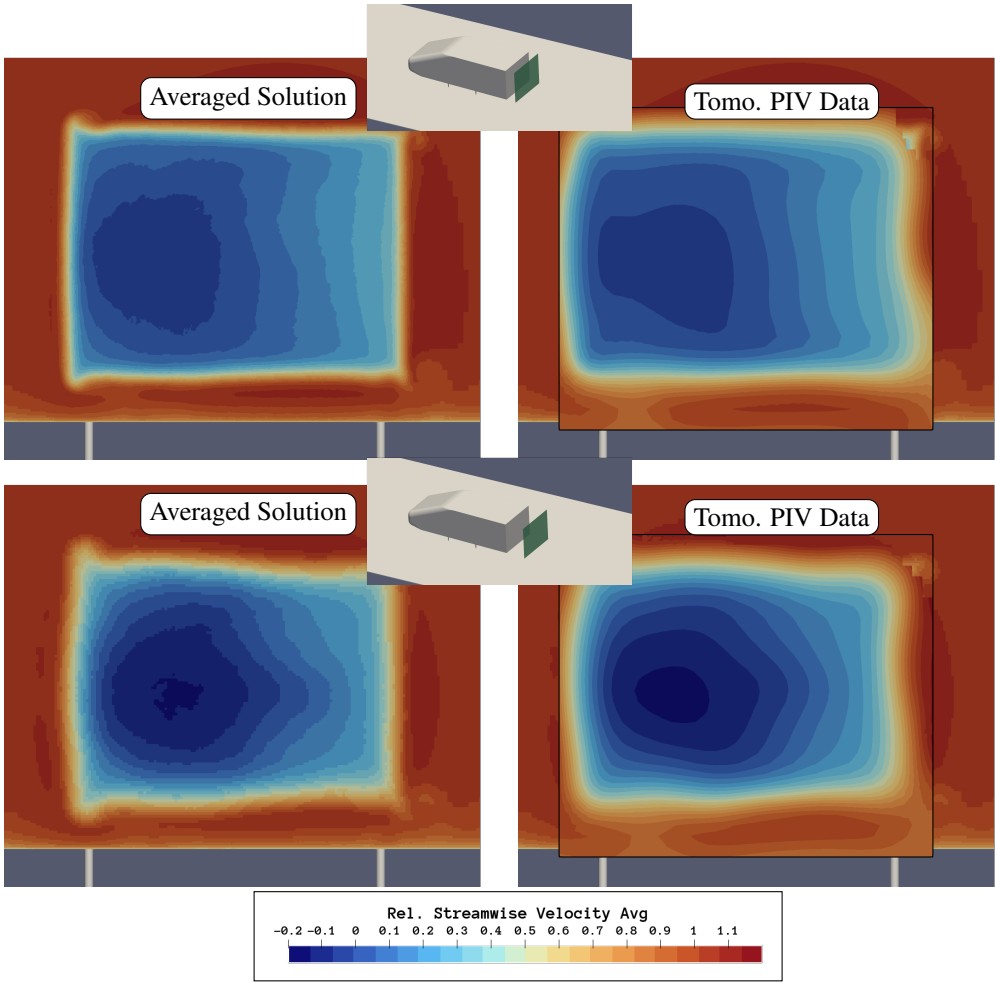

Figure 14: Time-averaged streamwise velocity at $x = 0.578\,\mathrm{m}$ (top) and $x = 0.670\,48\,\mathrm{m}$ (bottom).

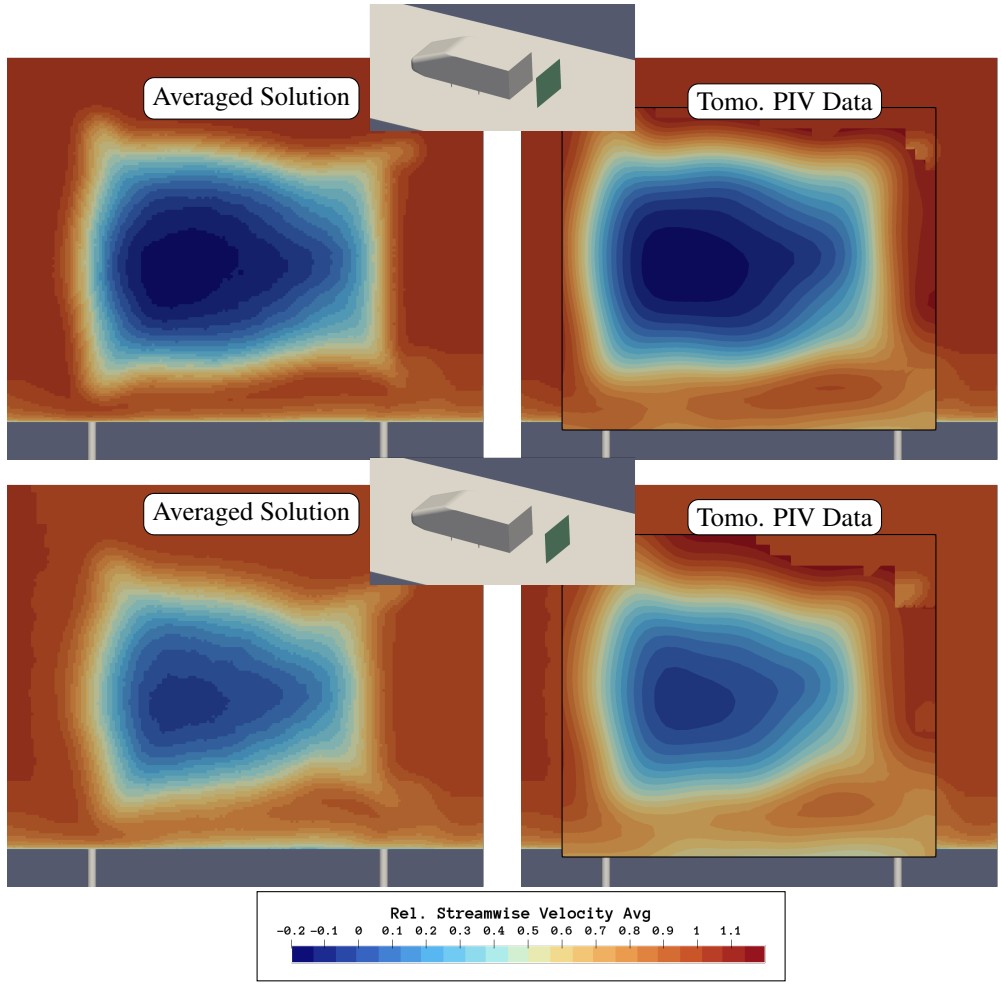

Figure 15: Time-averaged streamwise velocity at $x = 0.774\,52\,\mathrm{m}$ (top) and $x = 0.867\,\mathrm{m}$ (bottom).

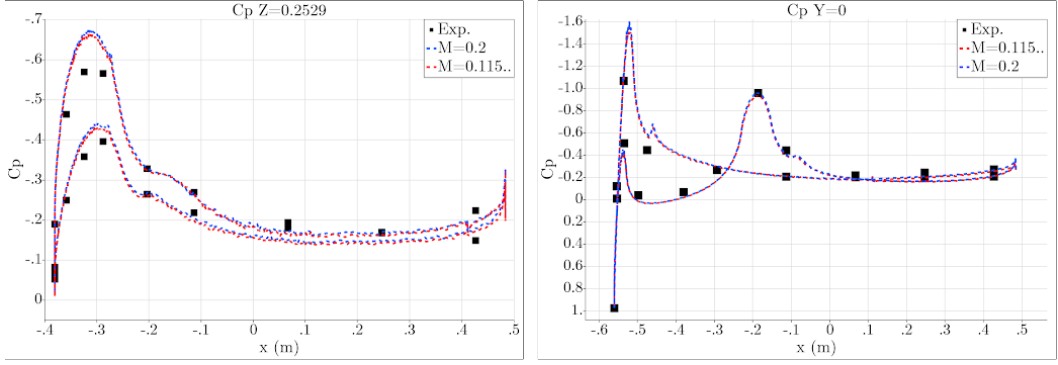

Figure 16: Comparison of pressure coefficient cuts on the Windsor body for two simulations on the same grid with different Mach numbers but the same Reynolds number. The solution is not degraded and the maximum stable time step is increased by 55% for the higher Mach number case.

# D ML evaluation

## D.1 Method

We have conducted preliminary analysis on our dataset using a modified version of one of the state-of-the-art scientific machine learning (SciML) methods, MeshGraphNets (6) on various tasks to illustrate the practicality of the dataset for ML evaluation. In both our method and MeshGraphNets, the node encoder and the edge encoder are 2-layer multilayer perceptrons (MLPs), and the processor consists of L message passing blocks, each containing two MLPs. There are a number of key differences between the approach taken in this work and that in the original paper of Pfaff et al.

The original justification and proof of MeshGraphNets was for transient prediction, which involves predicting dynamic quantities of the mesh at time t+1 given the current state and previous states. In contrast, our method focuses on directly predicting simulation results at the steady state or time-averaged state. We believe steady state or time-averaged prediction is more applicable in industrial contexts, where transient simulation data is often not preserved due to the large amount of data storage required.

This difference in task leads to distinct input node features. MeshGraphNet leverages dynamical features such as the instantaneous velocity or pressure to predict future states, but those features are not available in a steady state or time-averaged prediction. As our method predicts time-averaged results directly from meshes, the node features comprise node positions, node normals, and node defects in some use cases. Node positions represent the absolute node positions in the 3D space, as our method operates on 3D meshes. This is in contrast to the MeshGraphNets paper, which primarily focused on 2D CFD use cases. Our edge features, on the other hand, include the relative displacement vector (i.e., the difference between the positions of the two nodes forming the edge) and its norm, which closely resemble the edge features in MeshGraphNets.

Our model architecture has been adapted to address two distinct use cases: direct Key Performance Indictor (KPI) prediction and surface variable prediction. Among the two use cases, the architecture used in surface variable prediction more closely resembles that of MeshGraphNets, as both produce node level predictions. For KPI prediction, the models make graph level predictions, that is predicting one or a few values (e.g lift or drag coefficient) per graph rather than per node. To achieve this, there is a pooling layer being added between the processor and the decoder. The pooling layer converts node embeddings to graph embeddings which are then passed to the decoder.

## D.2 Results

The entire WindsorML dataset is split into training (60%), validation (20%), and test (20%) sets. For each use case, the model is trained on the training set, and the checkpoint that had the best validation error was used to obtain the inference results on the test set.

Using the first method to directly predict the lift and drag coefficients we find that using a 60/20/20 split of train, validation and test data, it is possible to obtain a MSE of less than 0.00028 for the drag coefficient and a MSE less than 0.0175 for the lift coefficient. An example of the prediction accuracy is shown in Figure 17 & 18 for the training, validation and test data for the lift and drag coefficient. Using x8 NVIDIA L40s GPUs (Amazon EC2 g6e.48xlarge instances via Amazon Web Services), the training completes in approximately 2hrs and the inference time for each new predicted geometry is 0.15 seconds. We ran this training for 1000 epochs using 5 message passing layers with a batch size of 4. The Adams optimizer was used with an initial learning rate of 0.003.

For the second method, where the lift and drag coefficient are obtained through the integration of the predicted surface pressure and wall-shear stress on the 4.4M node VTP surface mesh, we obtain predictions for the drag coefficient with a mean absolute error (MAE) of 0.03269 and a weighted mean absolute percentage error (WMAPE) of 0.1040. For the lift coefficient the mean absolute error (MAE) is 0.1854 and the weighted mean absolute percentage error (WMAPE) of 0.698 (as shown in Figures 19 & 20). Training time is approximately 60 hours on x8 NVIDIA L40s GPUs (Amazon EC2 g6e.48xlarge instances via Amazon Web Services) and the inference time is less than a minute on the same hardware. We ran this training with BF16 for 1200 epochs using 20 message passing layers with a batch size of 1. The Adams optimizer was used with an initial learning rate of 0.001.

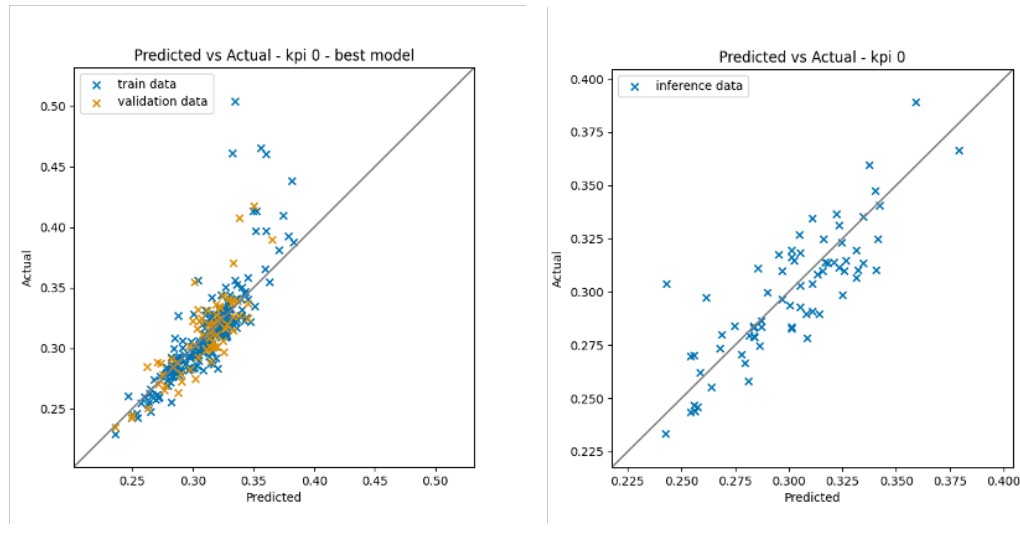

(a) Predicted drag coefficient on training/validation split (b) Predicted drag coefficient on test split

Figure 17: Prediction of the drag coefficient using the direct KPI method

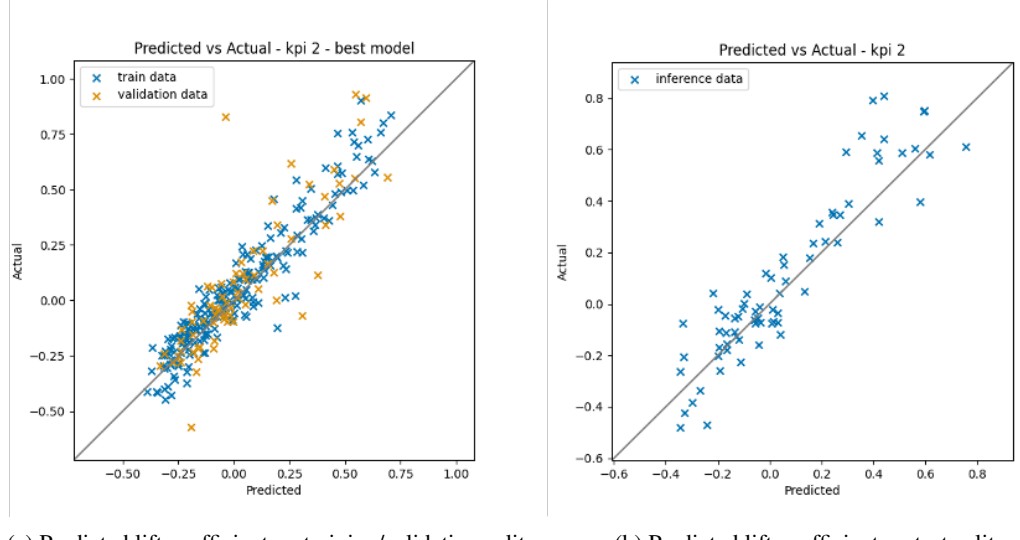

(a) Predicted lift coefficient on training/validation split (b) Predicted lift coefficient on test split

Figure 18: Prediction of the lift coefficient using the direct KPI method

It is clear from these results that the WindsorML dataset represents a challenging use-case for ML methods to predict, with noticeable difference in accuracy based upon the method used. Please note that these ML evaluations are preliminary and purely serve to illustrate how this dataset can be used for ML evaluation. We hope other groups will use this dataset to do a more thorough evaluation of different ML methodologies.

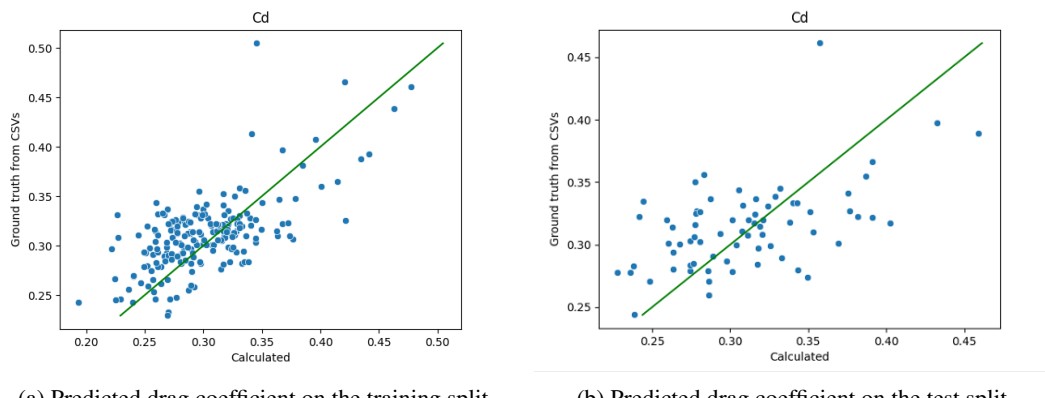

(a) Predicted drag coefficient on the training split   (b) Predicted drag coefficient on the test split

Figure 19: Prediction of the drag coefficient obtained through integration of the wall-shear stress and pressure

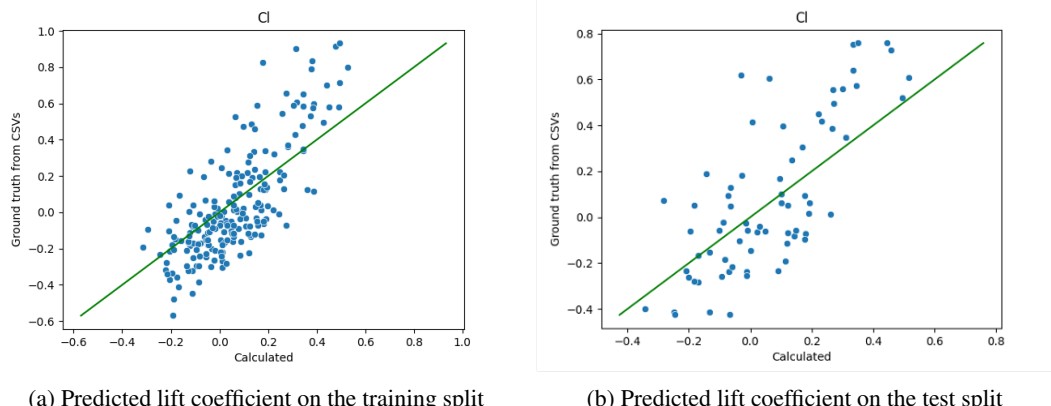

(a) Predicted lift coefficient on the training split   (b) Predicted lift coefficient on the test split

Figure 20: Prediction of the lift coefficient obtained through integration of the wall-shear stress and pressure

# E  Datasheet

## E.1  Motivation

- **For what purpose was the dataset created?** The dataset was created to address the current limitations of a lack of high-fidelity training data for the development and testing of machine learning methods for Computational Fluid Dynamics. In addition, it was created to be used as a dataset for the 4th Automotive CFD Prediction Workshop [3]

- **Who created the dataset (e.g., which team, research group) and on behalf of which entity (e.g., company, institution, organization)?** The dataset was created by a consortium of developers and scientists from academia and industry.

- **Who funded the creation of the dataset?** The project was internally funded within each author organisation i.e no external grants.

## E.2  Distribution

- **Will the dataset be distributed to third parties outside of the entity (e.g., company, institution, organization) on behalf of which the dataset was created?** Yes, the dataset is open to the public

- **How will the dataset will be distributed (e.g., tarball on website, API, GitHub)?** The dataset is free to download from Amazon S3 (without the need for an AWS account) and is fully described on the dataset website [4]. Additional download sites and options are in the process of being created and will be shared on the website once ready.

- **When will the dataset be distributed?** The dataset is already available to download via Amazon S3. Additional download sites and options are in the process of being created and will be shared on the website once ready.

- **Will the dataset be distributed under a copyright or other intellectual property (IP) license, and/or under applicable terms of use (ToU)?** The dataset is licensed under CC-BY-SA license.

- **Have any third parties imposed IP-based or other restrictions on the data associated with the instances?** No

- **Do any export controls or other regulatory restrictions apply to the dataset or to individual instances?** No

## E.3  Maintenance

- **Who will be supporting/hosting/maintaining the dataset?** The dataset is being managed by the collection of authors and a public website [5] will provide on-going updates on hosting/maintenance.

- **How can the owner/curator/manager of the dataset be contacted (e.g., email address)?** The owner/curator/manager of the dataset can be contacted at contact@caemldatasets.org (these are also provided in the dataset README and paper).

- **Is there an erratum?** No, but if we find errors we will provide updates to the dataset and note any changes in the dataset README and website.

- **Will the dataset be updated (e.g., to correct labeling errors, add new instances, delete instances)?** Yes the dataset will be updated to address errors or provided extra functionality. The README of the dataset and the dataset website will be updated to reflect this.

- **If the dataset relates to people, are there applicable limits on the retention of the data associated with the instances (e.g., were the individuals in question told that their data would be retained for a fixed period of time and then deleted)?** N/A

- **Will older versions of the dataset continue to be supported/hosted/maintained?** Yes, if there are substantial changes or additions, older versions will still be kept.

---

[3]https://autocfd.org

[4]https://caemldatasets.org

[5]https://caemldatasets.org

- **If others want to extend/augment/build on/contribute to the dataset, is there a mechanism for them to do so?** We will consider this on a case by case basis and they can contact contact@caemldatasets.org to discuss this further.

## E.4 Composition

- **What do the instances that comprise the dataset represent (e.g., documents, photos, people, countries)?** Computational Fluid Dynamics simulations.
- **How many instances are there in total (of each type, if appropriate)?** Table 1 details the specific outputs that are contained in the dataset for each of the 355 geometric variations of the Windsor body.
- **Does the dataset contain all possible instances or is it a sample (not necessarily random) of instances from a larger set?** The dataset is complete collection of simulations run to date.
- **What data does each instance consist of?** Table 1 details the specific outputs that are contained in the dataset for each of the 355 geometric variations of the Windsor body.
- **Is there a label or target associated with each instance?** N/A
- **Is any information missing from individual instances?** No, the dataset is fully described.
- **Are relationships between individual instances made explicit (e.g., users' movie ratings, social network links)?** N/A
- **Are there recommended data splits (e.g., training, development/validation, testing)?** 60/20/20 is the recommended split based upon initial testing.
- **Are there any errors, sources of noise, or redundancies in the dataset?** The errors associated with the Computational Fluid Dynamics method is discussed in the validation section of the main paper and SI.
- **Is the dataset self-contained, or does it link to or otherwise rely on external resources (e.g., websites, tweets, other datasets)?** It is self-contained.
- **Does the dataset contain data that might be considered confidential (e.g., data that is protected by legal privilege or by doctor patient confidentiality, data that includes the content of individuals' non-public communications)?** The data is fully open-source and not considered confidential.
- **Does the dataset contain data that, if viewed directly, might be offensive, insulting, threatening, or might otherwise cause anxiety?** No

## E.5 Collection Process

- **How was the data associated with each instance acquired?** The data was obtained through Computational Fluid Dynamics (CFD) simulations and then post-processed to extract only the required quantities.
- **What mechanisms or procedures were used to collect the data (e.g., hardware apparatus or sensor, manual human curation, software program, software API)?** The main paper discusses the HPC hardware used
- **If the dataset is a sample from a larger set, what was the sampling strategy (e.g., deterministic, probabilistic with specific sampling probabilities)?** N/A
- **Who was involved in the data collection process (e.g., students, crowdworkers, contractors) and how were they compensated (e.g., how much were crowdworkers paid)?** N/A
- **Over what timeframe was the data collected?** Simulation were run over the year of 2024.
- **Were any ethical review processes conducted (e.g., by an institutional review board)?** N/A

## E.6 Preprocessing/cleaning/labeling

- **Was any preprocessing/cleaning/labeling of the data done (e.g., discretization or bucketing, tokenization, part-of-speech tagging, SIFT feature extraction, removal of instances, processing of missing values)?** N/A

### E.7 Uses

- **Has the dataset been used for any tasks already?** Yes, limited testing with various ML approaches has been undertaken by the author team to ensure that the data provided in the dataset is suitable for ML training and inference.

- **Is there a repository that links to any or all papers or systems that use the dataset?** No

- **What (other) tasks could the dataset be used for?** The primary focus is for ML development and testing but it could also be used for the study of turbulent flows over bluff bodies.

- **Is there anything about the composition of the dataset or the way it was collected and preprocessed/cleaned/labeled that might impact future uses?** Not to the knowledge of the authors.

- **Are there tasks for which the dataset should not be used?** No