# OpenReview forum: "WindsorML: High-Fidelity Computational Fluid Dynamics Dataset For Automotive Aerodynamics"
_NeurIPS.cc/2024/Datasets_and_Benchmarks_Track — NeurIPS 2024 Track Datasets and Benchmarks Poster_

### Official Review · Reviewer_bbMf · 2024-06-24
**Review of WindsorML dataset**

**Rating:** 4
**Confidence:** 5
**Clarity:** The paper is well written

**Review:**

This paper introduces large eddy simulation aerodynamic dataset for machine learning task on car geometry. The dataset consists of 355 geometry variants of the baseline Windsor geometry. Wall-Modeled Large-Eddy Simulation was used to solve for the follow field on the car. The solver solution was tested for grid sensitivity and validation using PIV experimental data. The authors discussed the potential use of the dataset in developing and testing data-driven ML surrogate models. One of the main limitations of the dataset that it has single inflow condition as there are many low-fidelity datasets for external aerodynamics that cover wider parametric space. Additionally, the Windsor car geometry is not a realistic car geometry in comparison to the real car. So this approximation will introduce additional error to the ML model. Lastly, this work did not include any testing cases for existing ML models on the data.

**Strengths:**

1.	This work introduces a high fidelity LES dataset for accelerating the aerodynamic optimization in the automotive industry with machine learning.
2.	The dataset consists of the flow fields over 355 geometry variants of the Windsor body under single inflow condition.
3.	PIV experimental data was used to validate the CFD solver before generating the dataset. Additionally, the authors, performed mesh independence study to eliminate the solution sensitivity to grid size.

**Additional Feedback:**

No

**Correctness:**

Generally, the claims in the manuscript are correct. However, the paper needs further work to show a testing case of the dataset’s usage in ML training.

**Documentation:**

Check the “opportunities for improvement” for more details. Additionally, the maintenance plan of the dataset is not clear and whether it will be extended or not.

**Limitations:**

The authors should consider sustaining and extending the dataset with more configuration and inflow conditions to ensure the continuity and growth of this work. Single inflow condition is insufficient for performing aerodynamic optimization under real-world conditions.

**Opportunities For Improvement:**

1.	Proper documentation of the dataset on Amazon S3, along with a tutorial script for loading and accessing the data, is recommended. This would facilitate seamless integration and understanding by other researchers.
2.	The dataset was generated for a given inflow condition. Although the car geometry is crucial parameter in the automotive aerodynamics, different inflow conditions are still necessary to simulate all the possible scenarios in real world. Additionally, training a machine learning to predict aerodynamic parameters won’t be easy task due the underlying physics associated with shapes and the turbulent flow behavior.
3.	Since the Windsor car geometry is kind of approximated version of the real car exterior, it is missing a lot of the complexities that might contribute to the flow turbulence and hence affect the aerodynamic parameters. How could this dataset be used? and for what tasks? If there’s another realistic dataset as you claimed in the paper.
4.	Although, the dataset holds promising potential, it is crucial to validate the dataset and showcase its utilities by testing various machine learning models on it (not for benchmarking purposes). This would provide insights into the dataset's characteristics, its suitability for different tasks, and establish baseline performance metrics for comparison.
5.	The dataset looks not ready to be used as machine learning dataset. The current data is still saved in the filer format that is typically common in the CFD community. The authors are encouraged to explore options for converting VTK files into more commonly used formats in the ML datasets (e.g., NumPy arrays, HDF5, etc.). Whether the dataset will be used for regular or graph ML task, the dataset should be preprocessed and ready for integration in ML pipeline.
6.	The authors highlighted how this dataset differ from AhmedML and DrivAerML. Since all the datasets are proposing flow field simulations over variants of car geometry, they could have integrated all the datasets under one big dataset that will hold significant value than being divided in three papers.

**Relation To Prior Work:**

Yes, the authors clearly discussed the difference between this paper and the existing work in literature

**Summary And Contributions:**

This work presents high-fidelity LES flow field dataset over car geometries which holds  a good value for optimizing the car aerodynamics and geometry using machine learning. The dataset consists of 355 geometry variants of the Windsor geometry under single inflow conditions. The availability of such a high-fidelity dataset is crucial to advance ML research in automotive aerodynamics. However, this work still needs some work to have significant impact in the field.

---

> ### Author Rebuttal · Authors · 2024-08-17
>
> Thank you very much for your detailed review. We are pleased you see the strengths in this work "This work introduces a high fidelity LES dataset for accelerating the aerodynamic optimization in the automotive industry with machine learning.
> PIV experimental data was used to validate the CFD solver before generating the dataset. Additionally, the authors, performed mesh independence study to eliminate the solution sensitivity to grid size.", however we note a number of concerns, which we wanted to address in turn:
>
> **Proper documentation of the dataset on Amazon S3, along with a tutorial script for loading and accessing the data, is recommended. ...**
>
> We have included detailed instructions in the SI section including scripts on how to download the data, as well as in the dataset README.txt. We have also since created a website to help people more easily navigate the dataset contents (https://caemldatasets.org).
>
> **The dataset was generated for a given inflow condition. Although the car geometry is crucial parameter in the automotive aerodynamics, different inflow conditions are still necessary to simulate all the possible scenarios in real world...**
>
> Automotive aerodynamics is generally considered to be Reynolds number independent within the normal range of vehicle speeds, meaning that normalised aerodynamic coefficients show negligible dependence on variations in speed. In industry, the majority of design iterations are therefore conducted for a single inflow speed.
>
> We do however note your comment (and we mention it as a limitation in the paper) that future work would be to expand the dataset to consider additional inflow conditions (e.g different yaw angles to simulate cross winds). However, we believe that the current work is already a very important contribution to the community since it constitutes the first-ever attempt to create such a large dataset with high-fidelity CFD and thus we respectively believe we should not be overly penalized for not including different inflow conditions.
>
> **Since the Windsor car geometry is kind of approximated version of the real car exterior, it is missing a lot of the complexities that might contribute to the flow turbulence and hence affect the aerodynamic parameters. How could this dataset be used? and for what tasks? ...**
>
> As we discuss in the paper, the windsor body was created by automotive companies (Jaguar Land Rover) in collaboration with academia (Loughborough) to approximate the main flow physics from road-cars whilst keeping the advantage of geometry simplicity and a modifiable shape and components. The dominate flow physics contributing to the drag of a road vehicle is the flow separation from the rear of the car, which the windsor captures well. It is therefore well suited to sit in-between much simpler shapes and the complexity of a real-life car such as the DrivAer vehicle. The dataset will therefore be used to assess the capability of ML methods to predict the broad different types of flow physics from the vehicle, as most likely a stepping stone to doing the same for the more complex full road-car. We have discussed this in the paper but will do more to make it even clearer.
>
> **Although, the dataset holds promising potential, it is crucial to validate the dataset and showcase its utilities by testing various machine learning models on it (not for benchmarking purposes)....**
>
> We did not see any explicit requirement in the dataset/benchmark track to include ML evaluations and thus decided to purely focus on describing in detail the validation of the methods and the dataset contents and the process to generate it. However we accept your comment that it would be useful to have a section showing how this dataset can be used for ML evaluation and we have already trained a GNN approach (based upon MeshGraphNet - shown in earlier work by some of the authors using a different dataset [1]) using this data to predict the force coefficients directly as well as via the pressure and wall-shear from the surface mesh of the vehicle and we will include this in the paper and discuss other methods that people could use in a dedicated section.
>
> **The dataset looks not ready to be used as machine learning dataset. The current data is still saved in the filer format that is typically common in the CFD community. The authors are encouraged to explore options for converting VTK files into more commonly used formats in the ML datasets (e.g., NumPy arrays, HDF5, etc.)....**
>
> We respectfully disagree with your comment that most deep learning platforms not supporting .vtk and needing to provide the data in numPy arrays or HDF5. The purpose of the dataset is to prepare the ML community for the data formats that they would expect to get from real-life CFD outputs. For this reason, the majority of existing datasets in the CFD community ([2]) are provided in the .vtk/.vtu format and also now many open-source ML codes such as NVidia Modulus support .vtk formats natively as well as libraries such as triMesh and meshio. We will add a section to the paper outlining why we have chosen this format and we can add details of codes which can accept .vtk format and how you can use various libraries to read in .vtk.
>
> **The authors highlighted how this dataset differ from AhmedML and DrivAerML. Since all the datasets are proposing flow field simulations over variants of car geometry, they could have integrated all the datasets under one big dataset that will hold significant value than being divided in three papers.**
>
> We respectively disagree. Combining all three papers - whilst maintaining the same level of details on the validation, dataset contents (which has been noted as strengths by reviewers) would simply not be possible within the limits of a NeurIPS paper.
>
> [1] Ananthan, V., et al.,“Machine Learning for Road Vehicle Aerodynamics,” SAE, 2024.
> [2] Florent Bonnet, et al. AirfRANS: High Fidelity CFD Dataset for Approximating RANS Solutions. arxiv.org, 2022

---

> > ### Comment · Reviewer_bbMf · 2024-08-23
> > **Official Comment by Reviewer bbMf**
> >
> > Thank you for addressing some of the comments raised in the initial review.
> >
> > 1. **Model Geometry**: I still don’t see the benefits of prototyping a ML model on the Windsor body compared to the DrivAer model. While it’s true that the Windsor geometry was used by Land Rover to optimize aerodynamic performance, it’s important to note that this geometry was introduced before 2000 and lacks some of the modern design features found in current vehicles. In contrast, the DrivAer model, introduced by the TUM team in 2014, offers a more realistic and up-to-date representation of the aerodynamics. So developing a dataset based on a realistic geometry would likely yield more accurate and applicable results in real-world scenarios.
> > 2. **ML Implementation Examples**: As I previously mentioned, As I previously mentioned, this isn’t about benchmarking where you implement multiple models and conduct an in-depth analysis of the learning process. Instead, it’s an opportunity to showcase the practical application of the dataset with a few machine learning models while also characterizing the dataset and identifying any potential issues. If you’ve already trained a GNN model on a similar dataset, this task should be straightforward for you.
> > 3. **Dataset Format**: Yes, VTK/VTU format is currently used in Nvidia modulus for visualization which is based on paraview. Recently, they wrote a custom data loader module for Ahmed and DrivAer body datasets. However, it’s important not to limit the dataset’s usability to just NVIDIA Modulus pipeline.
> > 4. **small vs large dataset**: I understand your concern regarding the 9 pages limit before t. acknowledgements, bibliography. However, there is no page limit for appendices and supplementary material as long as the file size less than 50 MB. Please check the formatting instructions here https://neurips.cc/Conferences/2024/CallForPapers.

---

> > > ### Author Rebuttal · Authors · 2024-08-26
> > >
> > > Thank you for this continued discussion.
> > >
> > > 1. We believe both statements can be true. It is correct that there is strong value is creating a dataset on the most realistic openly available car geometry e.g DrivAer (which is work conducted in parallel by several of the co-authors [1]). However, we believe it is good scientific practice to assess any new approach on a range of test-cases, building up in complexity. The Windsor is well suited to be one of these stepping stone cases. In addition, the Windsor body has the advantage of being the standard test-case for the past two Automotive CFD Prediction Workshops [2] where there is a wealth of prior CFD simulation data. This will help developers who test on the Windsor to also consider how accurate traditional CFD approaches have been for this use-case.
> > >
> > > 2. We have responded to all the reviewers given it will most likely be of interest to all. Please see that rebuttal where we include details of the MeshGraphNet results. We are happy to provide any further details and of course we would include more details in a revised paper.
> > >
> > > 3. Similarly to point 1, we feel both points can be true i.e we strongly believe that a .vtk/vtu dataset makes sense to prepare the community for the sort of data that will come from actual CFD simulations - which is the target input data for the surrogate methods the community is developing. However it is also true that traditionally the ML community has preferred HDF5, NumPy arrays. As mentioned in previously comments, we will include details in the revised paper of how people can use common libraries to read in the .vtk/.vtu type formats within PyTorch etc.
> > >
> > > 4. We believe that each dataset has enough novelty to be reviewed and assessed independently. Whilst they were generated under the umbrella of a shared motivation, they were generated by different groups, using different codes so make sense to be included in separate papers. Whilst there is no limit to supplementary materials, the paper itself should be self-containing (as per NeurIPS guidelines i.e reviewers are not expect to have to even read them) and thus describing the results, codes and also ML evaluations (which will be included in a revised paper) in 9 pages will result in an unacceptable lost of quality and detail in our opinion.
> > >
> > > We thank you again for these discussions that very useful and we already have a number of things we would revise in the paper that I'm sure will enhance it's readability and quality thanks to your comments.
> > >
> > > [1] DrivAerML: High-Fidelity Computational Fluid Dynamics Dataset for Road-Car External Aerodynamics Neil Ashton, Charles Mockett, Marian Fuchs, and Louis Fliessbach, Hendrik Hetmann, Thilo Knacke, Norbert Schonwald, Vangelis Skaperdas, Grigoris Fotiadis, Astrid Walle, Burkhard Hupertz, Danielle Maddix arxiv.org, 2024, https://arxiv.org/abs/2408.11969
> > > [2] https://autocfd.org/

---

### Official Review · Reviewer_ZUyK · 2024-07-10
**Valuable CFD Benchmark Dataset**

**Rating:** 8
**Confidence:** 4
**Correctness:** The dataset construction is sound. Th…

**Review:**

This paper provides an open-source dataset that can be used for building and evaluating ML models that are used for CFD, specifically to evaluate the fluid flow around a vehicle. The dataset is large enough and diverse enough to support the proposed ML research. The model is developed using a newer vehicle model, more representative of current automotive designs. And the dataset has a sufficiently fine-grained mesh to capture detail that is needed for a comparison of ML models for more realistic applications.

CFD is a challenging problem for ML, as are all complex problems that require the solution of PDEs. Having a high-quality dataset will help researchers develop ML models that are more applicable to industry-grade problems.

The paper is well written with a logical organization and clear explanations of the underlying design decisions for the model, including the limitations of the dataset. The paper is original and provides a much better dataset for this problem an is available to the community elsewhere.

**Strengths:**

This paper is a significant advance in the benchmark datasets available for this problem and will be beneficial for the research community who is interested in solving CFD problems using ML.

There are no ethical implications but having accurate ML models that result in a faster solution of CFD problems would offer great benefits to society, such as more aerodynamic vehicles as well as represent a significant advance for ML as a discipline.

**Additional Feedback:**

None.

**Clarity:**

Yes, the paper is well written. The ideas are clear and there is sufficient description of the model. There are only a few minor edits needed. Specifically,

Page 2: "as well forces" should be "as well as forces".
Page 3: "whose flow features more closely..." should be written. It's probably correct but difficult to parse and understand, because "flow" could be a noun and "features" a verb, or "flow" could be an adjective modifying the noun "features".

There are numerous places where some abbreviations are given incorrectly, such as "e.g" instead of "e.g.," or "i.e" instead of "i.e.," or "etc" instead of "etc."

**Documentation:**

Yes, there is sufficient detail.

**Ethics:**

There are no ethical concerns with the dataset. It is created using artificial models that are designed to mimic realistic models, but doesn't include any personal information.

**Limitations:**

The authors have adequately addressed limitations of the work.

**Opportunities For Improvement:**

The paper describes the limitations of the current dataset in terms of its generality and what additional aspects could be explored in a more comprehensive dataset.

**Relation To Prior Work:**

Yes, the paper covers this sufficiently.

**Summary And Contributions:**

This paper describes a benchmark dataset for ML applied to CFD for automotive applications, specifically a wind tunnel simulation for turbulence learning. This dataset has some practical advantages over existing datasets for similar problems since it is of higher quality, so will provide a more realistic test for ML models for this very challenging problem.

The paper does not describe a ML model that uses this dataset and focuses solely on the benchmark dataset.

---

> ### Author Rebuttal · Authors · 2024-08-17
>
> Thank you very much for the detailed review. We are pleased you see the strengths of the paper and how it will be very useful to the CFD and ML communities. We will make sure to correct the mistakes you have identified in the clarity portion of your review to improve the flow of some of the sentences.

---

### Official Review · Reviewer_6P67 · 2024-07-23
**Impressive dataset but the lack of ML application in the paper is detrimental to its acceptance**

**Rating:** 5
**Confidence:** 5
**Correctness:** All the claims made are sound and sup…

**Review:**

The paper is extremely well written paper with a very detailed supplementary material.
The validation of the dataset shown in the main manuscript and supplementary material is rigorous.
The details given by the authors to access and use the dataset are also much appreciated.

My rating is explained by the two following issues:
- The usefulness of the dataset for industry is clear. However, I have some reserves about its use for research purposes. Providing some temporal outputs, as suggested by the authors as future work, would be extremely beneficial for the research community.
- All Neurips paper presenting fluid mechanics dataset that I have read display also a benchmark case, which is not the case of the current paper. A benchmark would be extremely useful to showcase the potential applications of the dataset for ML purposes.

The paper will deserve a bump up in the rating if the above points are tackled.

**Strengths:**

Very well written paper. Impressive dataset in its accuracy.

**Additional Feedback:**

N/A

**Clarity:**

There are some issues with the legibility of the figures.
- Figure 1: Font size too small, especially the axis.
- Figure 3: Font size too small.
- Figure 4 needs a colorbar.
- Figure 6: Font size too small.

Please clarify (potentially with added references) what *‘favorable Kinetic Energy and Entropy Consistency properties’* are.
Do the authors use shock capturing and/or limiters in the current simulations? If so, what is the typical fraction of the domain that needs such treatment?
A reference to the supplementary material where the CFD solver is discussed in detail should be provided in the main text.

**Documentation:**

The comprehensive documentation is much appreciated.

**Limitations:**

The dataset is obviously relevant for automotive aerodynamics, but this is a niche subject at the scale of ML research. However, the dataset in general has a much broader potential for fluid mechanics research. I ask again the authors to consider including temporal snapshots of the simulation in the dataset.

The authors partially motivate this new dataset by its more relevant application to automotive CFD compared to what is currently found in the literature. Could the authors discuss the differences and limitations arising from the boundary layer? Since such a boundary layer is not present when the car moves, it is my understanding that this might lead to some significant different physics.

**Opportunities For Improvement:**

I believe the intended use, described in the supplementary material, should also be discussed in the main manuscript. Perhaps even in more detail to guide future users about the strength and limitations of the dataset.

Please see further comments in the Review section.

**Relation To Prior Work:**

The literature discussion is acceptable for the context of automotive aerodynamics.

**Summary And Contributions:**

This paper presents a high-fidelity dataset for ML containing 355 geometric variants of an idealized automotive body

---

> ### Author Rebuttal · Authors · 2024-08-17
>
> Thank you very much for your review. We are pleased to see your comments such as "The paper is extremely well written paper with a very detailed supplementary material. The validation of the dataset shown in the main manuscript and supplementary material is rigorous. The details given by the authors to access and use the dataset are also much appreciated.". However we note your concerns, which we want to address in turn. Hope that by addressing your concerns you would consider a higher rating for this paper.
>
> **The usefulness of the dataset for industry is clear. However, I have some reserves about its use for research purposes. Providing some temporal outputs, as suggested by the authors as future work, would be extremely beneficial for the research community.**
>
> Thank for your comment. As a compromise we could include outputs per time-steps (not every single time-step) of the dataset for two geometries and then perhaps based upon community input we could decide how best to proceed from there. Generating time-accurate time-series for all geometries would be a huge increase in storage due to running more than 200,000 time-steps for each run, but we will explore how to provide a subset of the time-steps for some of the data outputs and include this in the updated paper.
>
> **All Neurips paper presenting fluid mechanics dataset that I have read display also a benchmark case, which is not the case of the current paper. A benchmark would be extremely useful to showcase the potential applications of the dataset for ML purposes.**
>
> We did not see any explicit requirement in the dataset/benchmark track to include ML evaluations and thus decided to purely focus on describing in detail the validation of the methods and the dataset contents and the process to generate it. However we accept your comment that it would be useful to have a section showing how this dataset can be used for ML evaluation and we have already trained a GNN approach (based upon MeshGraphNet - which was shown in earlier work by some of the authors using a different dataset [1]) using this data to predict the force coefficients directly as well as via the pressure and wall-shear from the surface mesh of the vehicle and we will include this in the paper and discuss other methods that people could use in a dedicated section. I hope this will address one of your main concerns.
>
> **Please clarify (potentially with added references) what ‘favorable Kinetic Energy and Entropy Consistency properties’ are. Do the authors use shock capturing and/or limiters in the current simulations? If so, what is the typical fraction of the domain that needs such treatment? A reference to the supplementary material where the CFD solver is discussed in detail should be provided in the main text.**
>
> We will add more details into the paper and make sure that some of the SI material is put in the main paper to help make things clearer. We will also add more details of the specific spatial discretizaion schemes to help answer that very question.
>
> **I believe the intended use, described in the supplementary material, should also be discussed in the main manuscript. Perhaps even in more detail to guide future users about the strength and limitations of the dataset.**
>
> We agree with your comment and will move this into the main paper. As per the previous comment, the addition of an example ML evaluation will further strengthen the intended use of the dataset.
>
> **The authors partially motivate this new dataset by its more relevant application to automotive CFD compared to what is currently found in the literature. Could the authors discuss the differences and limitations arising from the boundary layer? Since such a boundary layer is not present when the car moves, it is my understanding that this might lead to some significant different physics.**
>
> Excellent observation. We will add a paragraph to the paper discussing this. In general the moving of the floor is not as impactful as the rotating of the wheels (which of course are linked in reality). Firstly rotating the wheels adds extra uncertainty i.e how do you model the actual rotation of the wheels - via source terms, moving meshes etc. The tyres also in reality deform under load as they rotate. For these reasons it's very common for academic and industry R&D to remove this uncertainty when doing R&D, especially when one of the biggest contributions to overall drag is the separation from the rear-window. The ability to capture this large scale separation is a key challenge for CFD and why the Windsor body is very popular because you can assess the ability to capture the most dominate source of drag whilst removing the wheel/floor issue. So whilst the movement of the floor/wheels and the change in the boundary layer are important, the current configuration of the Windsor body that we use is still keen by academia and industry as very practically useful.
>
> **There are some issues with the legibility of the figures.**
> We will make sure to correct the issues with font sizes being too small.
>
> [1] Ananthan, V., et al.,“Machine Learning for Road Vehicle Aerodynamics,” SAE, 2024.

---

> > ### Comment · Reviewer_6P67 · 2024-08-19
> >
> > The rebuttal is satisfactory. At this stage, including temporal dataset for just 1 case seems reasonable. How many snapshots could be added?
> >
> > If the proposed changes (especially the inclusion of temporal dataset and the GNN results) are implemented to the manuscript and uploaded here before the final deadline, I will re-evaluate my rating.

---

> > > ### Author Rebuttal · Authors · 2024-08-23
> > >
> > > We have conducted early analysis on our dataset using a modified version of one of the state-of-the-art scientific machine learning (SciML) methods, MeshGraphNet (Pfaff et al., “Learning Mesh-Based Simulation with Graph Networks”, ICLR 2020) on various tasks. We have results on the time-averaged solution field to predict both the surface variables e.g pressure and wall-shear stress and also image prediction of the 2D wake flow-field slices on variables such as the Total Pressure Coefficient (CpT).  We also have results on the practical downstream task of predicting the force coefficients directly and also through the integration of the pressure and wall-shear stress. We utilize the encoder-processor-decoder architecture in MeshGraphNets and modify the method to enable it to make time-averaged predictions [Ananthan et al. Machine Learning for Road Vehicle Aerodynamics, SAE, 2024] .
> > >
> > > The dataset is split into training (60%), validation (20%), and test (20%) sets. For each use case, the model is trained on the training set, and the checkpoint that had the best validation error was used to obtain the inference results on the test set.
> > >
> > > For the Windsor dataset, using the predicted surface pressure and wall-shear stress on the downsampled surface mesh , we obtain predictions for the drag coefficient with a mean absolute percentage error (MAPE) of 0.135 and a mean absolute error (MAE): 0.042. Training time is approximately 5 hours on x8 NVidia A10g GPUs and the inference time is less than a minute on the same hardware. Please note that these runs are preliminary and further work to optimize the methodology and hyperparameters is on-going.
> > >
> > > These results show a proof of concept that these CFD datasets can be used to effectively train ML models

---

> > ### Author Response · Authors · 2024-08-28
> >
> > Please note that we have added the requested experiments with MeshGraphNets in the above rebuttal and pdf. As the review period is coming to a close, we hope that we have addressed your concerns and that you consider raising your score as you mentioned above. Thank you.

---

### Official Review · Reviewer_gnJc · 2024-07-24
**an open-source CFD dataset based on 355 geometry variants of the automotive Windsor body**

**Rating:** 8
**Confidence:** 3
**Correctness:** Yes.
**Clarity:** Yes.

**Review:**

Pros:
* high-fidelity and comprehensive dataset with 355 geometric variants.
* detailed validation against experimental data ensures reliability.
* well-organized and detailed presentation of methods and processes.
* unique and innovative dataset that could potentially advance research in automotive aerodynamics.

Cons:
* limited to geometric variations without boundary condition changes.
* represents an incremental improvement on existing datasets rather than a completely novel one.

**Strengths:**

Please see above.

**Additional Feedback:**

Please see above.

**Documentation:**

Yes.

**Ethics:**

No.

**Limitations:**

Yes.

**Opportunities For Improvement:**

Other surrogate models, such as GNNs, might benefit from access to velocity or pressure data at specific locations and time stamps. Efficiently enabling this capability would further enhance the dataset's utility. For instance, considering downsampling and open-sourcing the time series data could be very meaningful for applications that require the velocity field for visualization or other tasks that do not necessitate very high resolution.

**Relation To Prior Work:**

Yes.

**Summary And Contributions:**

This paper presents an open-source CFD dataset based on 355 geometry variants of the automotive Windsor body that exhibit a wide range of flow characteristics that are representative of those observed on road-cars. The dataset includes time-averaged volume & boundary data as well as the geometry and force & moment coefficients obtained from expensive large-scale fluid simulations. It is the first open-source dataset based on the windsor body that can be used to train surrogate models for automotive aerodynamic use cases.

---

> ### Author Rebuttal · Authors · 2024-08-16
>
> Thank you very much for your feedback. We appreciate your positive comments "high-fidelity and comprehensive dataset with 355 geometric variants; detailed validation against experimental data ensures reliability; well-organized and detailed presentation of methods and processes; unique and innovative dataset that could potentially advance research in automotive aerodynamics." however we want to address your main concerns:
>
> **Other surrogate models, such as GNNs, might benefit from access to velocity or pressure data at specific locations and time stamps. Efficiently enabling this capability would further enhance the dataset's utility. For instance, considering downsampling and open-sourcing the time series data could be very meaningful for applications that require the velocity field for visualization or other tasks that do not necessitate very high resolution.**
>
> This is a fair comment and something we are considering to do. However we wanted to first get feedback from users of this dataset and then consider how we can add and improve it. As you say one such approach would be to provide the time-series data (with downsampling to reduce the data size). This will not be possible for all geometries due to the very large size but it would be possible for a number of cases. We have this written down in the limitations section already and accept it is something that would be good to include in future iterations of this dataset.
>
> **limited to geometric variations without boundary condition changes.**
>
> We accept this is a limitation of this work (and hence include it in our limitations section).  However automotive aerodynamics is generally considered to be Reynolds number independent within the normal range of vehicle speeds, meaning that normalised aerodynamic coefficients show negligible dependence on variations in speed. It is also the case that typically in industry, the majority of design iterations are done for a constant boundary condition (given the geometry is the main change) however there are other boundary conditions e.g moving floor, or a yaw inflow angle change that are things we are considering for future iterations of the dataset. We accept it is a limitation but believe it is one that shouldn't be overly penalized given the novelty and scale of work to create even a constant boundary condition dataset of this scale.
>
> **represents an incremental improvement on existing datasets rather than a completely novel one.**
>
> It is true that other datasets have been released however none have been released for the Windsor body itself and none have used higher-fidelity methods such as WMLES in this work. So we believe it is still novel but as with much of science it is also technically also an incremental improvement.

---

### Author Rebuttal · Authors · 2024-08-23

We have conducted early analysis on our dataset using a modified version of one of the state-of-the-art scientific machine learning (SciML) methods, MeshGraphNet (Pfaff et al., “Learning Mesh-Based Simulation with Graph Networks”, ICLR 2020) on various tasks. We have results on the time-averaged solution field to predict both the surface variables e.g pressure and wall-shear stress and also image prediction of the 2D wake flow-field slices on variables such as the Total Pressure Coefficient (CpT). We also have results on the practical downstream task of predicting the force coefficients directly and also through the integration of the pressure and wall-shear stress. We utilize the encoder-processor-decoder architecture in MeshGraphNets and modify the method to enable it to make time-averaged predictions [Ananthan et al. Machine Learning for Road Vehicle Aerodynamics, SAE, 2024] .

The dataset is split into training (60%), validation (20%), and test (20%) sets. For each use case, the model is trained on the training set, and the checkpoint that had the best validation error was used to obtain the inference results on the test set.

For the Windsor dataset, using the predicted surface pressure and wall-shear stress on the downsampled surface mesh , we obtain predictions for the drag coefficient with a mean absolute percentage error (MAPE) of 0.135 and a mean absolute error (MAE): 0.042. Training time is approximately 5 hours on x8 NVidia A10g GPUs and the inference time is less than a minute on the same hardware. Please note that these runs are preliminary and further work to optimize the methodology and hyperparameters is on-going.

These results show a proof of concept that these CFD datasets can be used to effectively train ML models

---

### Decision · Program_Chairs · 2024-09-26

**Decision:**

Accept (Poster)

**Comment:**

The paper presents a unique and non-conventional open-source dataset that for the first time offering a benchmarking option that can be utilized for systematic evaluation of a broad range of Physics-Informed ML models as well as algorithms from Computational Fluid Dynamics (CFD), on the basis of automotive aerodynamic use cases. While the paper has received polarized reviews and a number of important questions has been identified that need to be addressed such as automatic data quality control, it appears that the strengths of the paper outweigh weaknesses and the paper has a high potential for creating novel synergy between the ML and scientific computing communities.